# Comparison of the multifractal characteristics of heavy metals in soils within two areas of contrasting economic activities in China

Xiaohui Li[a, b], Xiangling Li[a], Feng Yuan[a, b*], Simon M. Jowitt[c,d], Taofa Zhou[a], Kui Yang[a], Jie Zhou[a], Xunyu Hu[a], Yang Li[a]

a. School of Resources and Environmental Engineering, Hefei University of Technology, Hefei 230009, China

b. Xinjiang Research Centre for Mineral Resources, Xinjiang Institute of Ecology and Geography, Chinese Academy of Sciences, Urumqi, Xinjiang 830011, China

c. School of Earth, Atmosphere and Environment, Monash University, Wellington Road, Clayton, VIC 3800, Australia

d. Department of Geoscience, University of Nevada Las Vegas, 4505 S. Maryland Parkway, Las Vegas, NV 89154-4010, USA

*Corresponding author: Email: yf_hfut@163.com, Tel: +8605512901648

**Abstract**

Industrial and agricultural activities can generate heavy metal pollution that can cause a number of negative environmental and health impacts. This means that evaluating heavy metal pollution and identifying the sources of these pollutants, especially in urban or developed areas, is an important first step in mitigating the effects of these contaminating but necessary economic activities. Here, we present the results of a heavy metal (Cu, Pb, Zn, Cd, As and Hg) soil geochemical survey in Hefei city and use a multifractal spectral technique to identify and compare the multifractality of heavy metal concentrations of soils within the industrial Daxing and agricultural Yicheng areas. This paper uses three multifractal parameters ($\Delta\alpha$, $\Delta f(\alpha)$ and $\tau''(1)$) for these soil geochemical data to indicate the overall amount of

multifractality within the soil geochemical data. The results show all of the elements barring Hg have larger $\Delta\alpha$, $\Delta f(\alpha)$ and $\tau''(1)$ values in the Daxing area compared to the Yicheng area. The differences in the degree of multifractality between Daxing and Yicheng areas indicate that the soils in these areas have differing multifractal geochemical characteristics, suggesting that the differing economic activities in these areas generate very different heavy metal pollutant loads. In addition, the industrial Daxing area contains significant Pb and Cd soil contamination, whereas Hg is the main heavy metal present in soils within the Yicheng area, indicating that differing clean-up procedures and approaches to remediating these polluted areas are needed. The results also indicate that multifractal modeling and the associated generation of multifractal parameters can be a useful approach in the evaluation of heavy metal pollution in soils.

**Keywords:** soil geochemistry; multifractal modelling; heavy metal pollution; Hefei

## 1. Introduction and overview of the study area

Heavy metal pollution within soil poses a serious risk for human health and the environment, and thus soil pollution caused by anthropogenic activities (including industry and agriculture) has been the focus of a significant amount of research (e.g., Leyval et al., 1997; Thomas and Stefan, 2002; McGrath et al., 2004; Wang et al., 2007; Luo et al., 2011). Analyzing soil geochemistry and pollution using multifractal techniques can investigate many of the problems of nonlinear variability which commonly arise when dealing with pollutants and as well as enabling the identification of non-linear characteristics within datasets. This approach can yield new information that can be used to understand the factors controlling the distribution of key elements within the objects or data being studied (Salvadori, 1997; Gonçalves, 2000; Zuo et al., 2012). This in turn means that determining the multifractal characteristics of the distribution of heavy metals in soils can improve our understanding of any heavy metal pollution that is associated with these differing

anthropogenic activities.
Multifractal techniques include singularity mapping and multifractal
interpolation that enable more detailed analysis of the spatial distribution of heavy
metals, concentration-area modeling that can be used to define threshold values
between background (i.e. geological) and anthropogenic anomalies (Lima et al., 2003),
spectral density-area modeling that can be used to define thresholds to separate
anomalies (i.e., anthropogenically derived heavy metal concentrations in this case)
from background concentrations (i.e., geologically derived heavy metal
concentrations; Cheng, 2001), and multifractal spectra that highlights non-linear
characteristics and identifies anomalous behavior that reflects the characteristics of
some multifractal sets (Gonçalves, 2000; Albanese et al., 2007; Guillén et al., 2011),
such as the presence of porous structures and spatial variations in soil properties
(Caniego et al., 2005; Dathe et al., 2006). This means that multifractal techniques can
be useful tools for the analysis of heavy metal pollution within soils (e.g., Salvadori et
al., 1997; Lima et al., 2003; Albanese et al., 2007; Guillén et al., 2011). These
multifractal techniques are not only used in environmental science, but also in a
number of differing fields, including geophysics (Schertzer et al., 2011), medicine
(Jennane et al., 2001), computer science (Wendt et al., 2009), geology (Cheng, 1995;
Deng et al., 2011; Zuo et al., 2012, 2014; Yuan et al., 2012, 2015; Nazarpour et al.,
2014) and ecology (Scheuring and Riedi, 1994; Pascual et al., 1995), among others.
Hefei is the capital of Anhui Province, China, and has an urban area that includes
the towns of Daxing and Yicheng, which focus on industrial and agricultural activities,
respectively. Here, we use multifractal spectra techniques and three parameters ($\Delta\alpha$,
$\Delta f(\alpha)$ and $\tau''(1)$) to analyze and compare the degree and characteristics of the
multifractality of heavy metal contamination in soils associated with anthropogenic
activities in this region. The results will further enable and inform future planning for
any necessary remediation of the soils in the Daxing and Yicheng areas.
## 2. Study area and geochemical data
### 2.1 Study area
The city of Hefei is situated in central–eastern China (Fig. 1(a)), has
approximately 7.7 million inhabitants and covers an area of around 11,408 km$^2$. This
paper focuses on the towns of Daxing and Yicheng (Fig. 1(b)), with the former
representing one of the traditional industrial areas of Hefei and containing numerous
factories that are involved in the steel industry, chemical industry, paper making, and
the production of furniture and construction materials, among others. In contrast, the
town of Yicheng focuses its economic activities on agricultural production, byproduct
processing, livestock and poultry breeding, ornamentals, and other enterprises related
to agricultural activity.
**2.2 Sampling and analysis**
The study areas are covered by Quaternary sedimentary soils and are free of both
natural mineralization and mining-related contamination. A total of 169 surface (<20
cm depth) soil samples were taken from the towns of Daxing and Yicheng on 1 × 1
km grids, yielding 78 samples from Daxing and 91 samples from Yicheng (Fig.
1(c–d)). Sampling errors were minimized by splitting each sample into 3–5
sub-samples, each of which weighed more than 500 g. Each of these sub-samples was
air-dried before being broken up using a wooden roller and then sieved to pass
through a 0.85 mm mesh. The concentrations of 6 heavy metal elements (Cu, Pb, Zn,
Cd, As and Hg) were determined during this study, with Cd, Cu, Pb and Zn
concentrations determined by inductively coupled plasma–mass spectrometry
(ICP–MS), whereas Hg and As concentrations were determined by hydride
generation–atomic fluorescence spectrometry (AFS; Armstrong et al., 1999;
Gómez-Ariza et al., 2000). These techniques have detection limits of 1 ppm for Cu, 2
ppm for Pb and Zn, 30 ppb for Cd, 0.5 ppm for As and 5 ppb for Hg. The accuracy of
these data was monitored by repeat and replicate determinations using instrumental
neutron activation analysis (INAA), with analytical precision was monitored using
variance of the results obtained from duplicate analyses.

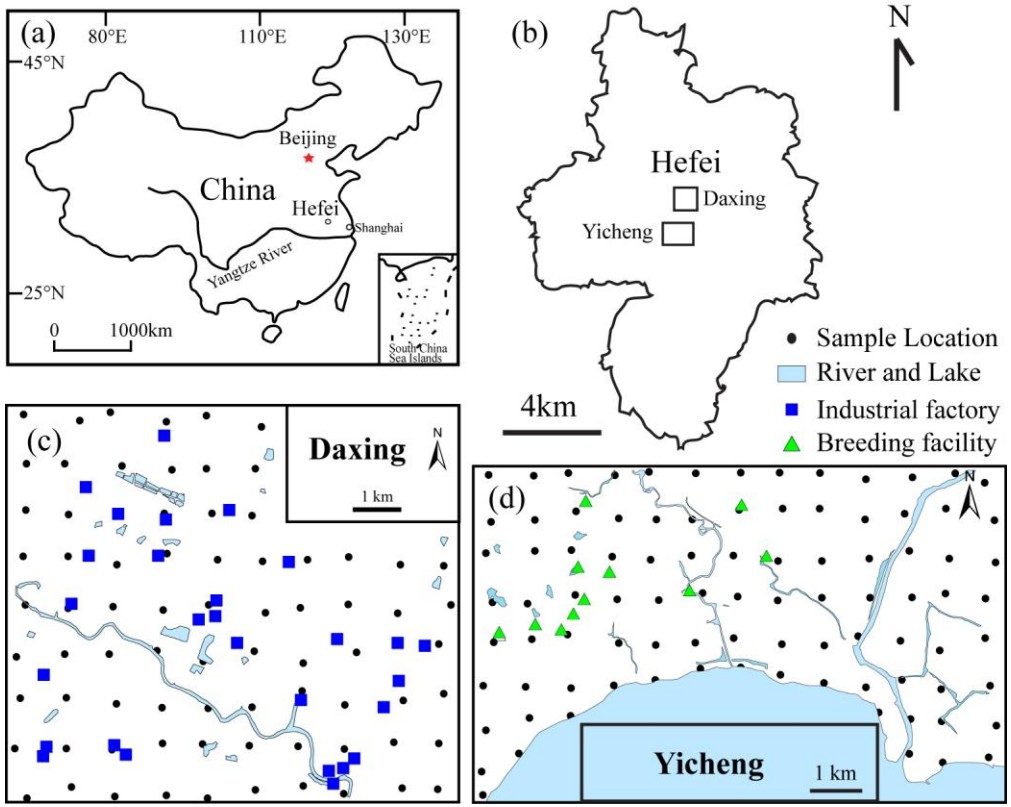

Fig. 1. Location of Hefei in central-eastern China (a); location of the study areas within Hefei (b); the 1 × 1 km grids used for soil sampling in the towns of Daxing (c) and Yicheng (d)

## 3. Multifractal spectrum analysis

Multifractal formalisms can decompose self-similar measures into intertwined fractal sets that are characterized by singularity strength and fractal dimensions (Cheng, 1999). Using multifractal techniques allows non-linear characteristics within datasets to be identified, enabling the extraction of information that can be used to understand the factors controlling the distribution of key elements within the data. Fractal spectra ($f(\alpha)$) are formalisms that can be used to describe the multifractal characteristics of a dataset and can be estimated using box-counting based moment, gliding box, histogram and wavelet methods, among others (Cheng, 1999; Lopes and Betrouni, 2009). The most widely used of these methods are the box-counting and gliding box methods, both of which are based on the moment method.

The calculation of the mass exponent function $\tau(q)$ for the gliding box method is different from the box-counting method, with the gliding box method providing a

useful approach that can increase the number of samples that are available for
statistical estimation within a dataset (Buczkowski et al., 1998; Tarquis et al., 2006;
Xie et al., 2010). This means that the gliding box approach often provides better
results with lower uncertainties than the box-counting method (Cheng, 1999). As such,
we have used the gliding box approach during this study. The calculation of the mass
exponent function $\tau(q)$ for the gliding box method uses a partition function as follows
(Cheng, 1999):
$$\langle \tau(q) \rangle + D = \lim_{\varepsilon \to 0} \left( \frac{\log(\mu^q(\varepsilon))}{\log(\varepsilon)} \right) = \lim_{\varepsilon \to 0} \left( \frac{\log \left( \frac{1}{N^*(\varepsilon)} \right) \sum_{i=1}^{N^*(\varepsilon)} \mu_i^q(\varepsilon)}{\log(\varepsilon)} \right) \quad (1)$$
where $\mu_i(\varepsilon)$ denotes a measure with the $i_{th}$ cell of a gliding box of size $\varepsilon$, $q$ is the order
moment of this measure, $\langle \rangle$ indicates the statistical moment, and $N^*(\varepsilon)$ indicates the
total number of gliding boxes of size $\varepsilon$ with $\mu_i(\varepsilon)$ values different from 0.
The values of $\tau(q)$ derived using this equation can be then used to determine $a$
and $f(\alpha)$ values using a Legendre transformation, as expressed below:
$$\alpha(q) = \frac{d\tau(q)}{dq} \quad (2)$$
$$f(\alpha) = q\alpha(q) - \tau(q) = q\frac{d\tau(q)}{dq} - \tau(q) \quad (3)$$
$\Delta\alpha$ and $\Delta f$ are essential parameters required to analyze the multifractal
characteristics of the dataset in question. The widths of the left and right branches
within the multifractal spectra are then defined using the following equations:
$$\Delta\alpha_L = \alpha_0 - \alpha_{min} \quad (4)$$
$$\Delta\alpha_R = \alpha_{max} - \alpha_0 \quad (5)$$
$$\Delta\alpha = \alpha_{max} - \alpha_{min} \quad (6)$$
and the height difference $\Delta f(\alpha)$ between the two ends of the multifractal spectrum is
then extracted using:
$$\Delta f(\alpha) = f(\alpha_{max}) - f(\alpha_{min}) \quad (7)$$

Higher $\Delta\alpha$ and $\Delta f(\alpha)$ values are generally indicative of datasets with more heterogeneous patterns (ordered, complex, clustered) and higher levels of multifractality (Cheng, 1999; Kravchenko et al., 1999). In addition, local multifractality $\tau''(1)$, can also be used as a measure to quantitatively characterize the multifractality of a dataset using equation 8, where ordinary spatial analysis functions (autocorrelations and semivariograms) are related to low order statistical moments (0 to 2nd) that may determine $\tau''(1)$ (Cheng, 2006):

$$\tau''(1) = \tau(2) - 2\tau(1) + \tau(0) \tag{8}$$

If $\mu$ is a multifractal and $-D < \tau''(1) < 0$, where D is the gliding-box dimension, then more negative values of $\tau''(1)$ are indicative of higher degrees of multifractality, whereas otherwise $\tau''(1) = 0$ for a single fractal.

Here, we use the three multifractal parameters described above ($\Delta\alpha$, $\Delta f(\alpha)$ and $\tau''(1)$) to better identify the degrees of multifractality within the soil geochemical data for the study area as well as enabling the comparison of the multifractality of differing elements in the soils in this region.

## 4. Geochemical analysis results

A statistical summary of the soil geochemical data for the study area is given in Table 1. Samples from the Daxing area have higher Cu, Pb, Zn, Cd and As maximum, mean, standard deviation, skewness, and kurtosis values than soil samples from the Yicheng area, whereas the Yicheng area has a higher maximum Hg concentration value than the Daxing area. In addition, the soil samples from Daxing have much higher coefficient of variation (CV) values for Cu, Pb, Zn, Cd and As than the samples from the Yicheng area, indicating that soils in the Daxing area contain higher and more variable concentrations of these elements. This also suggests that samples from the Daxing area containing elevated concentrations of heavy metals were probably contaminated by anthropogenic activity.

All of the elements (barring Pb and Cu in the Yicheng area) in both the Yicheng and Daxing areas yielded concentration histograms that are positively skewed and contain some outliers (Fig. 2), indicating that these data have non-normal and

potentially fractal- or multifractal-type distributions. This means that multifractal
techniques are highly suited for the characterization of the geochemistry of the soils
and discrimination of the differing types of human activities ongoing in each area.

Table 1. Summary statistics of soil heavy metal concentrations within samples from the Daxing
and Yicheng areas.

| Town | Element | Min | Max | Mean | Standard deviation | Skewness | Kurtosis | CV* |
|------|---------|-----|-----|------|--------------------|----------|----------|-----|
| | | (mg/kg) | (mg/kg) | (mg/kg) | - | - | - | (%) |
| Daxing | Cu | 19.00 | 111.50 | 33.87 | 13.26 | 3.20 | 14.93 | 39.16 |
| | Pb | 18.90 | 291.30 | 39.57 | 35.03 | 5.37 | 35.41 | 88.51 |
| | Zn | 40.90 | 526.10 | 105.8 | 94.40 | 2.91 | 8.59 | 89.19 |
| | Cd | 0.045 | 1.48 | 0.23 | 0.24 | 3.45 | 13.81 | 108.23 |
| | As | 4.93 | 308.20 | 13.97 | 33.89 | 8.72 | 76.64 | 242.56 |
| | Hg | 0.03 | 0.60 | 0.11 | 0.11 | 2.68 | 7.78 | 107.29 |
| Yicheng | Cu | 9.60 | 37.80 | 24.34 | 5.77 | -0.38 | 0.41 | 23.71 |
| | Pb | 10.40 | 46.30 | 22.77 | 4.91 | 0.87 | 5.51 | 21.56 |
| | Zn | 20.80 | 194.80 | 54.70 | 21.43 | 3.45 | 20.27 | 39.17 |
| | Cd | 0.054 | 0.43 | 0.15 | 0.08 | 1.84 | 3.49 | 51.85 |
| | As | 2.30 | 44.20 | 7.29 | 4.39 | 6.68 | 56.55 | 60.24 |
| | Hg | 0.02 | 0.62 | 0.06 | 0.07 | 5.75 | 41.26 | 113.09 |

*CV: coefficient of variation.

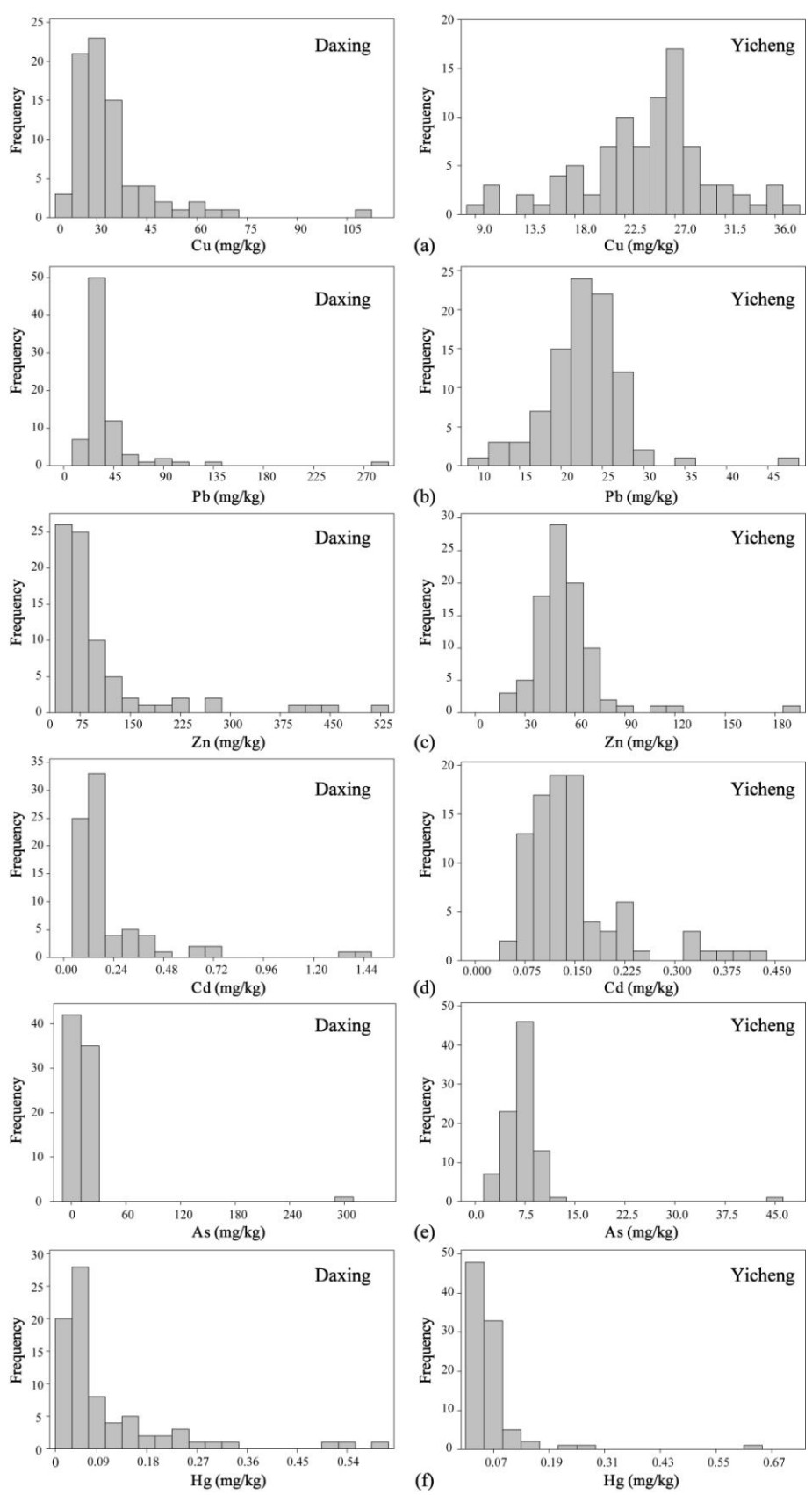


**Fig. 2.** Histograms showing the distribution of Cu (a), Pb (b), Zn (c), Cd (d), As (e) and Hg (f)
concentrations within soils from the towns of Daxing and Yicheng.

## 5. Calculation processes of multifractal spectrum and discussion

The multifractal spectra (in the form of an $\alpha$–$f(\alpha)$ diagram) for the geochemical data are shown in Fig. 3 using a range of $q$ values from −10 to 10 with an interval of 1.

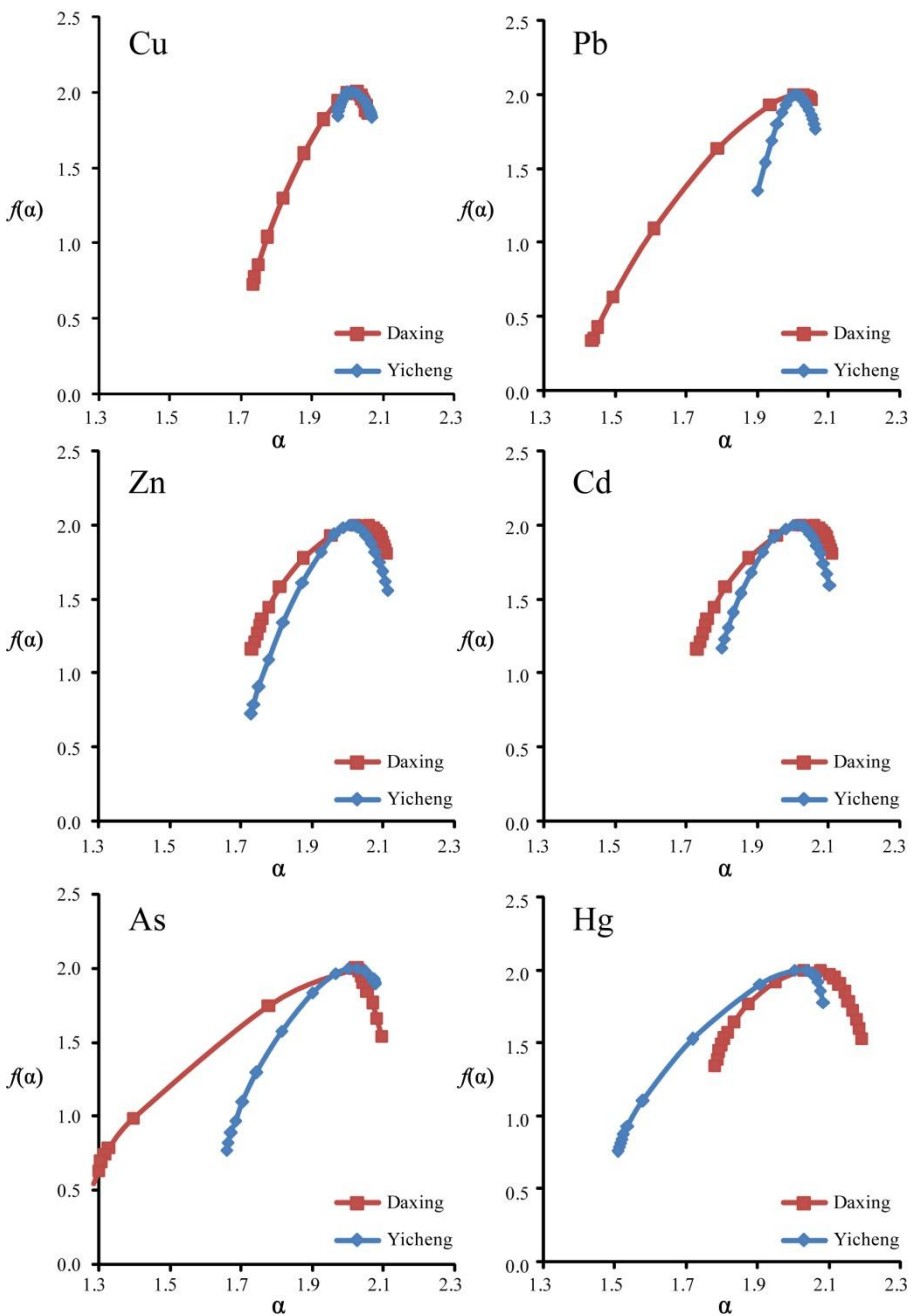

**Fig. 3.** Multifractal spectra ($f(\alpha)$ vs $\alpha$) of the soil geochemical data from the Daxing and Yichen area.

These multifractal spectra have inverse bell shapes (Fig. 3) and are asymmetric

(i.e. $\Delta\alpha_L$ values significantly differ from $\Delta\alpha_R$, equations 5-6) with the exception of the
Cu data for soils from the Yicheng area, indicating that the samples containing low
and high concentrations of these elements are not evenly distributed within the study
area (as is expected for areas containing point source pollutants like factories or
animal breeding facilities).

The multifractal results given in Table 2 indicate that all of the elements (barring

Cu and Pb in the Yicheng area) are characterized by a wide range of $\alpha$ values with
$\tau''(1)$ values less than –0.01 and $\Delta f(\alpha)$ values larger than 0.5, all of which indicate that
these elements have highly multifractality within the soils in these two areas. All of
the elements analyzed during this study (barring Hg) have higher $\Delta f(\alpha)$ and $\alpha$ values
(except Zn) and lower $\tau''(1)$ values in soils from the Daxing area, with Hg having
higher $\Delta f(\alpha)$ and $\alpha$ and lower $\tau''(1)$ values in soils from the Yicheng area (Table 2).
This suggests that the industrial activities in the Daxing area generate multi-element
heavy metal soil contamination, whereas the most significant heavy metal pollution
associated with the agricultural activity in the Yicheng area is Hg contamination. The
$\Delta f(\alpha)$ and $\alpha$ values of Hg in the Yicheng area are larger than the values for all other
elements in this area as well as some of the elements in the Daxing area, indicating
both the prevalence and significant degree of agricultural Hg contamination in the
Yicheng area, even considering the lower overall (but not maximum) concentrations
of Hg within the Yicheng area compared to the Daxing area. This contamination
should be considered a priority in terms of remediation, because the interaction
between the agricultural activity in the Yicheng area and this Hg pollution
could seriously impact human health, as Hg is preferentially concentrated upward in
the food chain (e.g. (Jiang et al., 2006)). This means that although contamination in
both areas needs to be evaluated further and should be remediated to avoid any
deleterious effects such as the heavy metal pollution of people, crops and animals, the
fact that the Hg contamination in the Yicheng area may be more bioavailable and may
have a larger effect on the population of this region (as a result of the agricultural
activity in this area) means it should be considered a priority.

**Table 2.** Multifractal parameters of the elements analyzed during this study.

| Town | Element | $\alpha_{min}$ | $\alpha_{max}$ | $\Delta\alpha_L$ | $\Delta\alpha_R$ | $\Delta\alpha$ | $\Delta f(\alpha)$ | $\tau''(1)$ |
|------|---------|----------------|----------------|------------------|------------------|----------------|--------------------|-------------|
| Daxing | Cu | 1.733 | 2.057 | 0.280 | 0.044 | 0.324 | 1.270 | -0.015 |
| | Pb | 1.439 | 2.050 | 0.567 | 0.044 | 0.611 | 1.659 | -0.068 |
| | Zn | 1.733 | 2.109 | 0.288 | 0.088 | 0.376 | 0.841 | -0.066 |
| | Cd | 1.482 | 2.285 | 0.499 | 0.304 | 0.803 | 1.358 | -0.066 |
| | As | 1.285 | 2.094 | 0.739 | 0.070 | 0.809 | 1.490 | -0.243 |
| | Hg | 1.780 | 2.191 | 0.248 | 0.163 | 0.411 | 0.656 | -0.079 |
| Yicheng | Cu | 1.971 | 2.067 | 0.036 | 0.060 | 0.096 | 0.168 | -0.007 |
| | Pb | 1.900 | 2.062 | 0.104 | 0.058 | 0.162 | 0.646 | -0.005 |
| | Zn | 1.729 | 2.112 | 0.275 | 0.108 | 0.383 | 1.275 | -0.016 |
| | Cd | 1.800 | 2.103 | 0.201 | 0.102 | 0.303 | 0.829 | -0.023 |
| | As | 1.659 | 2.076 | 0.343 | 0.075 | 0.418 | 1.224 | -0.036 |
| | Hg | 1.507 | 2.084 | 0.497 | 0.080 | 0.577 | 1.243 | -0.096 |

In order to compare variations in multifractality, different elements within Daxing and Yicheng area were sorted by $\Delta\alpha$, $\Delta f(\alpha)$ and $\tau''(1)$ parameters, respectively, in addition to sorting by basic statistics such as standard deviation and coefficient of variation values (Table 3). The data shown in Table 3 indicates that the Zn data within the Daxing area has largest standard deviation value but only a moderate coefficient of variation, but the $\Delta\alpha$ and $\Delta f(\alpha)$ values for these Zn data are indicative of only weak multifractality compared to the other heavy metals in the soils within the Daxing area. In comparison, the Hg data for soils in the Yicheng area yielded the lowest standard deviation value but the largest $\Delta\alpha$ and $\tau''(1)$ values, indicating these Hg data have strong multifractality. These differences indicate that the multifractal parameters $\Delta\alpha$, $\Delta f(\alpha)$ and $\tau''(1)$ reveal new information about the nonlinear variability and the characteristics of these geochemical data compared to the basic statistics for these samples. In addition, the data given in Table 3 indicates that these elements have different orders depending on whether they are sorted by $\Delta\alpha$, $\Delta f(\alpha)$ or by $\tau''(1)$ values, all of which reflects differing aspects of the multifractality of these data. Here we consider that $\Delta\alpha$, $\Delta f(\alpha)$ or by $\tau''(1)$ have equal weightings that reflect the overall multifractality of the data from the study area. As such, the ordering of these elements by $\Delta\alpha$, $\Delta f(\alpha)$ or by $\tau''(1)$ involved the summation of these values with the summed ordering then sorted again to compare the overall multifractality of these data.


<p style="text-align:center">**Table 3.** Elements sorted by multifractal parameters and basic statistic indices.</p>

| Town | Element | Order | | | | |
|---|---|---|---|---|---|---|
| | | Basic statistics | Multifractal parameters | | | |
| | | Coefficient of variation | $\Delta\alpha$ | $\Delta f(\alpha)$ | $\tau''(1)$ | Overall* |
| **Daxing** | Cu | 6 | 6 | 4 | 6 | 6 |
| | Pb | 5 | 3 | 1 | 1 | 1 |
| | Zn | 4 | 5 | 5 | 2 | 4 |
| | Cd | 2 | 2 | 3 | 3 | 2 |
| | As | 1 | 1 | 2 | 5 | 3 |
| | Hg | 3 | 4 | 6 | 4 | 5 |
| **Yicheng** | Cu | 5 | 6 | 6 | 5 | 6 |
| | Pb | 6 | 5 | 5 | 6 | 5 |
| | Zn | 4 | 3 | 1 | 4 | 3 |
| | Cd | 3 | 4 | 4 | 3 | 4 |
| | As | 2 | 2 | 3 | 2 | 2 |
| | Hg | 1 | 1 | 2 | 1 | 1 |

Overall: the overall order of $\Delta\alpha$, $\Delta f(\alpha)$ and $\tau''(1)$.

The overall amount of multifractality within the soil geochemical data for the
Daxing area decreases as follows: Pb>Cd>As>Zn>Hg>Cu, whereas the overall
amount of multifractality within the soil geochemical data for the Yicheng area
decreases as follows: Hg>As>Zn>Cd>Pb>Cu. The overall orders indicates that the Pb
and Hg soil data have the highest degree of multifractality in the Daxing and Yicheng
areas, respectively, whereas Cu has the weakest multifractality irrespective of the
area.
We further analyzed the spatial distribution of contamination within soils from
the Daxing and Yicheng areas and evaluated whether there is any significant
correlation between multifractality and anthropogenic activity. Filled contour maps
showing the distribution of Pb in the Daxing area and Hg and Cu in the Yicheng area
were calculated using inverse distance weighted interpolation (Fig. 4–6). These
figures show that areas with elevated levels of Pb contamination within the Daxing
area are correlated with the location of industrial factories, although interestingly the
areas in the upper and lower left hand side of Fig. 4 contain factories but not elevated
concentrations of Pb. This indicates that the Pb concentrations in these soils may be
dependent on both the presence and type of industry in this area, with some industries
more polluting than others, either as a direct result of the differing industries present
in this area or as a result of differing (or a lack of in some areas) approaches to
lessening environmental impacts. In comparison, the Hg contamination in the Yicheng
area is definitely spatially correlated with the location of agricultural breeding
facilities. Although the mean concentrations of Hg in soils are greater in the Daxing
area, all of the multifractal parameters determined during this study ($\Delta\alpha$, $\Delta f(\alpha)$ and
$\tau''(1)$) indicate that the Hg data in the Daxing area has a lower multifracticality than
the Hg data in the Yicheng area. The Yicheng area is heavily agricultural, meaning
that the agricultural activities in this area may be both concentrating Hg as well as
contaminating soils. In addition, although the mean concentrations of Hg in the
Yicheng area are lower than in the soils in the Daxing area, the former has a higher
maximum concentration than the latter, and both areas have significant Hg
contamination. Indeed, the contamination in the Yicheng area may be of more concern
than the contamination in the Daxing area, as the agricultural activity in the Yicheng
area may lead to greater human intake of Hg than from the soils in the mainly
industrial Daxing area, a factor that could lead to serious health issues (e.g. Minamata
disease) caused by the potential concentration of Hg up the food chain. This indicates
that soils in both areas may well require control and remediation.
This distribution of soils with elevated concentrations of Hg also contrasts with
the symmetrical distribution and weakest multifractality for Cu within the Yicheng
area (Fig. 3, 5-6). Here, we generated a correlation matrix that compares the
relationship between the spatial density of breeding locations in the Yicheng area (Fig.
7) and filled contours maps showing the distribution of Hg (Fig. 5) and Cu (Fig. 6) in
this region to identify whether there are any spatial correlations between the location
of agricultural facilities and areas containing soils with elevated heavy metal
concentrations (Table 4). The correlation matrix shows a significant correlation
between agricultural facilities and high concentrations of Hg (correlation value =
0.434), whereas the location of these agricultural breeding facilities and areas of high
Cu concentrations either have no relationship or are negatively correlated (correlation
value = –0.064). This indicates that very little Cu has been anthropogenically added
(or removed) from the soils in the Yicheng area, suggesting that these soils may
contain only natural background concentrations of Cu and that the breeding facilities
in this area does not produce significant Cu contamination. The correlation matrix,
symmetrical distribution and weakest multifractality for Cu give one clue to the
derivation of the Cu contamination in this area is the spatial relationship between Cu
contamination and the river in the right hand side of Fig. 6. This may suggest a
non-anthropogenic source (e.g. flooding causing the deposition of Cu or some other
relationship between water and Cu contamination) for some of the slightly elevated
Cu concentrations in this region. In addition, the fact that some breeding facilities are
not associated with significant Hg contamination (Fig. 5) suggests again that although
there is a relationship between the presence of these facilities and contamination, it
may be that the Hg contamination in this area reflects differing types of breeding
facilities or differing (or a lack of) approaches to lessening environmental impacts.
These results indicate that multifractal modeling and the associated generation of
multifractal parameters are a useful approach in the evaluation of heavy metal
pollution in soils and the identification of major element of heavy metal
contamination. In addition, the differing orders of the multifractality of the
geochemical data for soils within the Daxing area and Yicheng area are indicative of a
significant difference in the geochemical characteristics (and heavy metal pollution)
in the soils within these two areas. This indicates that differing treatment strategy and
clean-up approaches to remediating these two polluted areas are needed, rather than a
single cover-all strategy and approach to the remediation of heavy metal pollution. A
significant number of different remediation approaches can be used to resolve the
issues of heavy metal soil contamination (e.g., Bech et al., 2014; Koptsik, 2014).
Although somewhat beyond the scope of this study, the multi-element nature of the
contamination in the Daxing area means that physical and chemical approaches to
remediation (i.e., soil removal, soil vitrification, soil consolidation, electroremediation,
or soil washing) are probably well suited for the remediation of heavy metal
contaminated soil in this region (especially Pb). In comparison, the differing (i.e.
Hg-dominated) type of soil contamination in the Yicheng area could be more
efficiently treated using microremediation and phytoremediation, primarily as the
agriculture in this area requires a rapid reduction in the mobility and biological
availability of heavy metals in the soils (Mulligan et al., 2001; Wang et al., 2006). In
addition, the source of the Hg contamination (e.g. fertilizer, fodder, pesticides, water,
or some other source remains unclear. Identifying this source is also beyond the scope
of this paper although it is also clearly an area for future research, as the identification
of the source or sources of this contamination may prevent the future heavy metal
pollution of soils in this region.

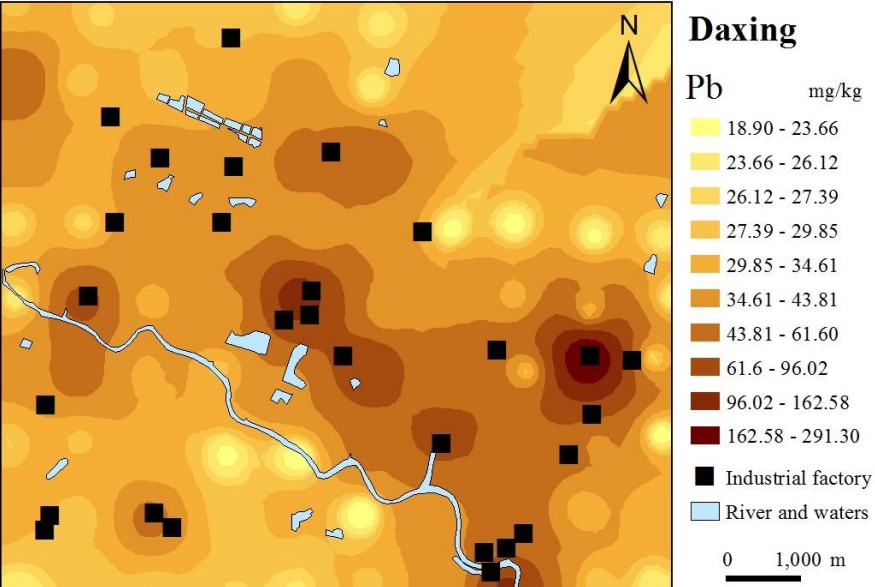


**Fig. 4.** Filled contour map generated by inverse distance weighted interpolation showing the
spatial distribution of soil Pb concentrations in the Daxing area.


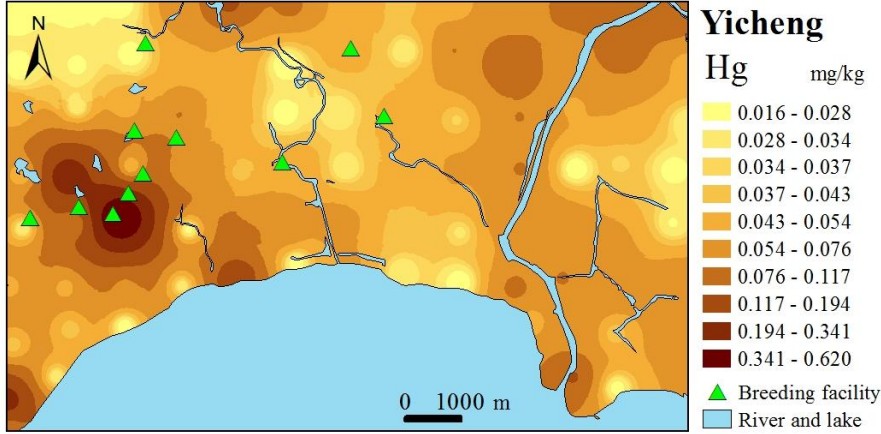

**Fig. 5.** Filled contour map generated by inverse distance weighted interpolation showing the
spatial distribution of soil Hg concentrations in the Yicheng area.

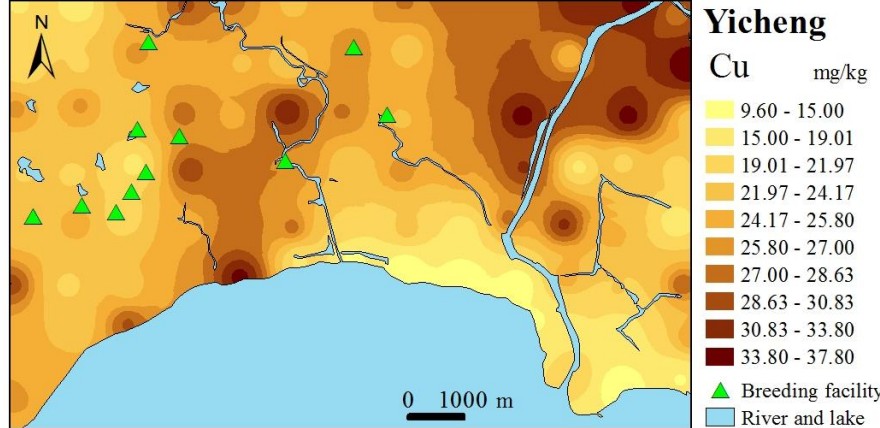

**Fig. 6.** Filled contour map generated by inverse distance weighted interpolation showing the
spatial distribution of soil Cu concentrations and the location of breeding facilities in the Yicheng
area

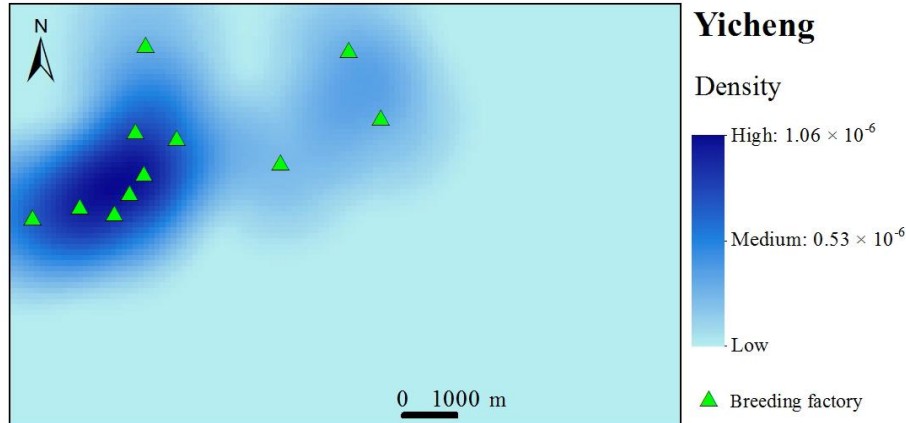

**Fig. 7.** Density map of breeding facilities in Yicheng area (generated using the kernel density
tool within the ArcGIS software package).
**Table 4.** Correlation matrix comparing the breeding facility density map and the filled

contour maps for Hg and Cu data for the Yicheng area.

| Layers | Layer 1 | Layer 2 | Layer 3 |
|--------|---------|---------|---------|
| Layer 1 | 1.00000 | 0.434 | –0.064 |
| Layer 2 | 0.434 | 1.000 | –0.464 |
| Layer 3 | –0.064 | –0.464 | 1.000 |

Layer 1: Density map of breeding factories of Yicheng area (Fig. 8);

Layer 2: Filled contour map of Hg concentrations of Yicheng area (Fig. 8);

Layer 3: Filled contour map of Cu concentrations of Yicheng area (Fig. 8).

The correlations range from +1 to –1, where a positive correlation indicates a direct relationship between the two layers and a negative correlation means that one variable is negatively correlated with the other. A correlation of zero means that two layers are independent of one another.

## 5. Conclusions

This study focuses on the geochemistry of heavy metal contaminated soils from the Daxing and Yicheng areas, both of which are located close to the city of Hefei, in Anhui Province, China. Multifractal modelling and the resulting multifractal parameters indicate that the soils from the Daxing area have stronger multifractality for Cu, Pb, Zn, Cd and As than soils from the Yicheng area, although the latter have relatively strong multifractality for Hg. The ordering of values for the multifractal parameters $\Delta\alpha$, $\Delta f(\alpha)$ and $\tau''(1)$ indicate the degree of multifractality for the geochemical data for soils within the Daxing area descends as follows: Pb>Cd>As>Zn>Hg>Cu, whereas the overall order in soils within the Yicheng area descends as follows: Hg>As>Zn>Cd>Pb>Cu. In addition, Cu concentrations in soils in the Yicheng area may still have their original (i.e. natural) distribution and may not have been influenced by human activities. These data indicate that the industrial activity concentrated in the Daxing area generates multi-element heavy metal soil contamination whereas the agricultural activity concentrated in the Yicheng area generates Hg-dominated heavy metal soil contamination. The latter is important, as Hg contamination can cause serious health issues (e.g. Minamata disease) and the soils in this area may well require remediation, especially as Hg can be concentrated up the food chain and the Yicheng area is heavily agricultural, indicating that this activity may both be concentrating Hg as well as contaminating soils in this area.

The results presented here indicate that multifractal modeling and the associated three multifractal parameters ($\Delta\alpha$, $\Delta f(\alpha)$ and $\tau''(1)$) can efficiently reflect the multifractality caused by industrial and agricultural activities in the Daxing and Yicheng areas, respectively. This in turn indicates that multifractal modeling can be a useful approach in the evaluation of heavy metal pollution in soils and the identification of problematic heavy metals that need remediation in the research area.

## Acknowledgements

This research was financially supported by funds from the China Academy of Science "Light of West China" Program, the Fundamental Research Funds for the Central Universities, and the Programme for New Century Excellent Talents in University (Grant No. NCET-10-0324).

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
