# Peer review of "metals in soils within two areas of contrasting economic"

_Nonlinear Processes in Geophysics, 2016_

## Referee Comment (RC1) · Anonymous Referee #1 · 15 Mar 2016

This small note describes a good experimental dataset collected in two contracted landuse areas. The manuscript has to be amended with hypothetical explanation of processes causing the observed multifractal properties. Otherwise it is purely descriptive narrative that does add to science. The recommendations of soil reclamation have to be substantiated with suggestions of reclamation techniques and purposes of reclamation, otherwise these recommendations are superfluous. Note that references are mostly old, all older than 5 years.

---

## Referee Comment (RC2) · J. Miranda (Referee) · 19 Mar 2016

GENERAL COMMENTS

Interesting article that uses the multifractal spectrum to assess possible soil contamination by industrial and agricultural activity in two regions of China. An excellent data collection work and proper use of the chosen methods. The proposed use of the distribution of alpha singularity exponents to evaluate diffusion of contaminants in the soil is valid, but the conclusions require more robust criteria of causality. The main conclusions outlined by the authors are based on correlations and comparisons that were not carefully evaluated. The authors use visual and inaccurate comparisons to validate important statements in the paper argument. The following is a detailed description of

suggestions to improve the work.

SPECIFIC COMMENTS

Line 138: I would say it suggests a non-normal distribution a priori, only. The possible fractal/multifractal pattern is something to be evaluated a posteriori.

210: Why did you choose these values of q? Is there any argument (e.g., when the Dqxq curve stabilises)?

Line 227: A comparison between the $\Delta f(\alpha)$ of the locations is considered here. The authors claim significant differences comparing only the order of the metals, sorted by $\Delta f(\alpha)$. Here a paired comparison statistic could prove the significant difference between areas.

232,234: In my point of view, Figure 3 shows no sufficient evidence to conclude about correlations between the spectra of the two regions. A correlation test between $\Delta\alpha$ (left and right) in Daxing and $\Delta\alpha$ (left and right) in Yicheng could give more support to the argument.

255-257: A logistic correlation could substantiate the statement of significant correlation between the location of industrial/agricultural facilities and metals concentrations.

258-260: A very interesting hypothesis, associating the asymmetry of the spectra with the presence of anthropic actions. Whereas the single symmetric spectrum found was the Cu in the Yicheng area, we would expect a not significant logistics correlation between the presence of agricultural facilities and concentration of Cu in Yicheng (map in Figure 6) and significant correlation in the asymmetry cases. These tests would substantiate the argument of using multifractal for evaluation of anthropogenic changes.

TECHNICAL COMMENTS

143: Just a suggestion: Make the legend a bit clear. The legend information is spread in the figure.

197: I would say more heterogeneous patterns, given the non-binary feature of heterogeneity.

214: "that describing the multifractality" - Unnecessary text.

221: The f spectrum is only another way to characterise your set. I am not sure if 'best measure' is the most suitable term.

234: Asymmetry concept could be better explained, it is presented in a way which might lead to misunderstandings. I would suggest an explanation based on the equations of the lines 191, 192 and 193.

238: Just two missed commas – "All of the heavy metals analyzed during this study, barring Hg, have higher $\Delta f(\alpha)$ values in soils from the Daxing area, with Hg having higher values in soils from the Yicheng area (Table 2)."

241: "The only significant heavy metal pollution associated with the agricultural activity in the Yicheng area WOULD BE the Hg contamination"

---

## Editor Comment (EC1) · J. M. Miras Avalos (Editor) · 21 Mar 2016

**The manuscript entitled "Multifractal characteristic-based comparison of elements in soils within the Daxing and Yicheng areas of Hefei, Anhui Province, China" (Reference number NPG-2016-15) authored by X. Li, F. Yuan, S.M. Jowitt, X. Li, T. Zhou, J. Zhou, X. Hu, and Y. Li presents results from a survey focused on heavy metals in two areas with contrasting economic activities (one area is industrial, the other one is agricultural). Authors applied the multifractal formalism to achieve indices that discern between both datasets.**

**The reported work is interesting and fits perfectly well within the scope of the Special Issue "Multifractal analysis in soil systems" to be published in Nonlinear Processes in Geophysics. However, the manuscript is not properly organized, English must be greatly improved and the discussion is very weak and biased to what happened on the agricultural area, which may suggest that it is the most contaminated one, which is not. Besides, there are some mistakes in the results.**

**In the following lines, I provide the authors with some suggestions in order to improve their manuscript. Therefore, the authors must address these issues prior to the acceptance of their manuscript. They must correct them in order that this manuscript achieves the standard quality for being published in *Nonlinear Processes in Geophysics*.**

**Therefore, I recommend a major revision prior to its publication in this Special Issue.**

**Specific comments to the authors:**

*Please, organize the manuscript in Introduction, Materials and Methods, Results and Discussion, and Conclusions.*

*Title:*

*I suggest the authors to slightly modify the title of their manuscript to: "Comparison of the multifractal characteristics of heavy metals in soils within two areas of contrasting economic activities in China".*

*Abstract:*

*The abstract is too long from my point of view.*

*Lines 16-17: "causing" instead of "that can have".*

*Line 23; "allows deeper interrogation", this expression is not clear. Please, re-phrase it.*

*Lines 26-28: "This study focuses…", this sentence can be removed since its information is reported in the next one.*

*Line 29: Include "(industrial)" after "Daxing" and "(agricultural)" after "Yicheng".*

*Lines 31, 32 and 38: Use $\alpha$ instead of a in $\Delta f(\alpha)$, please.*

*Line 33: There is a mistake here; according to table 2, the $\Delta f(\alpha)$ in Yicheng decreased as $Zn>Hg>As>Cd>Pb>Cu$ instead of $Hg>Zn>As>Cd>Pb>Cu$ as is reported here.*

*Line 34: I would remove the word "geochemical".*

*Line 36: "clearly different" instead of "distinctly different".*

*Lines 44-45: I would remove "rather than a single approach to heavy metal pollution" since it is not needed.*

*Introduction:*

*This section is not clear, the state-of-the-art is not put into context and thus the introduction seems out of focus. Moreover, this section begins with a list of references because in the first 6 lines, authors cited 13 references.*

*Line 51: I would remove "recently".*

*Lines 58-59: "the factors controlling the distribution" instead of "the controls on the distribution".*

*Line 62: "in soil properties" instead of "in the characteristics of soils".*

*Line 63: Remove "and".*

*Lines 63-65: Please, check English, this sentence is unclear.*

*Lines 67-68: "but also in the analysis of" instead of "but can also be used in the analysis of".*

*Line 70: "and thus" instead of "meaning that".*

*Lines 73-75: Please, re-phrase this sentence. It is not clear what you mean and must be put in context with the former sentence.*

*Line 77: Please, define "C-A" when first used.*

*Line 79: Please, define "S-A" when first used.*

*Lines 78-83: This is not clear, please, revise it.*

*Line 86: Remove "provincial".*

*Line 87: Remove "areas".*

*Line 88: "activities" instead of "activity".*

*Lines 88-96: This portion of the text is repetitive and unclear. Please, revise it and state clearly the aims of your study.*

*Study area and geochemical data:*

*Line 99: Include "it" before "has".*

*Line 102: "industrial areas of Hefei" instead of "industrial bases of the Hefei area".*

*Line 103: Remove the word "industrial".*

*Line 105-106: "In contrast, the town of Yicheng focuses its economic activities on agricultural production" instead of "In contrast, the town of Yicheng is agricultural, with the economy of the town focused on agricultural production".*

*Line 107: "ornamentals" instead of "flower planting".*

*Line 110: I do not understand what you mean by "natural mineralization".*

*Lines 110-111: "(< 20 cm depth)" instead of "(<20 cm below surface)".*

*Line 114: "was air-dried" instead of "was dried in air".*

*Line 117: Remove "in the soil samples described above", remove also "during this study".*

*Line 119: "whereas Hg and As concentrations were determined" instead of ""with Hg and As concentrations determined".*

*Lines 122-125: You repeat too many times the word "analysis", sometimes you can use the synonym "determinations".*

*Lines 116-125: Have you got references for the analytical methods? If so, please, add them to this portion of the text.*

*Line 126: "2.3. Results", this should be a section after the explanations of the materials and methods used.*

*Line 127: "A statistical summary" instead of "The results of a statistical analysis".*

*Line 134: "the natural background". Maybe you should indicate what was the natural background.*

*Line 136: I would include Pb with Cu for the Yicheng area since the distribution of its concentrations in soils seems to follow a normal distribution as well.*

*Line 137: I would include "(Fig. 2)" after "outliers".*

*Line 138: I am not sure, I agree that they are non-normal but how can you tell from the histograms that they are fractal?*

*Line140: Remove "(Fig. 2)" from here.*

*Lines 143-146: I would rephrase this figure caption to "Location of Hefei in central-eastern China (a); location of the study areas within Hefei (b); 1 x 1 km grid for soil sampling in the towns of Daxin (c) and Yicheng (d)".*

*Line 148: Re-phrase the title of this table to "Summary statistics of soil heavy metal concentrations from the Daxing and Yicheng samples".*

*Table 1: Skewness and kurtosis are not concentrations and they are dimensionless. I would put the*

*units below each column title, I mean below "minimum", "maximum", "mean" and "standard deviation". I would remove "Concentrations" from the table.*

*Mutifractal spectrum analysis:*

*Equations should be numbered.*

*Line 159: "the factors controlling the distribution" instead of "the controls on the distribution". What do you mean by "of key elements within data".*

*Line 160: Remove the word "multifractal". "f($\alpha$)" instead of "f(a)".*

*Line 163: Remove "of estimating f(a) values" since it is not needed.*

*Line 170: "different from 0" instead of "that $\neq$ 0".*

*Lines 173-174: Move "within a dataset" to after "statistical estimation".*

*Line 183: "different from 0" instead of "that $\neq$ 0".*

*Lines 184-185: Use alpha ($\alpha$) instead of a when referring to the multifractal spectra.*

*Line 194: "spectrum is" instead of "spectrum are".*

*Line 197: Use alpha ($\alpha$) instead of a when referring to the multifractal spectra.*

*Line 201: "by the following" instead of "using the following".*

*Calculation processes and discussion:*

*Lines 205-209: This has already been said in the former section.*

*Line 214: Remove "that" and "all of".*

*Lines 217-223: This description should be greatly improved. Check English, please. Only Cua and Pb for Yicheng area have $\tau''(1)$ values lower than -0.01.*

*Line 225: Use "indices" instead of "elements". You are not talking about the elements but the multifractal indices that you obtained.*

*Line 226: "decrease" instead of "decreases". There is a mistake here, Zn have a greater f($\alpha$) value than Hg for the Yicheng samples.*

*Line 227: This has already been observed in the statistical summary.*

*Line 229-231: In fact, you are plotting these data.*

*Lines 232-234: This is not clear. Please, re-phrase it.*

*Line 235: Remove "for all elements".*

*Lines 235-240: I am not sure about understanding this. Please, re-phrase it.*

*Line 241: "heavy metal contamination of soil" instead of "heavy metal contamination soil contamination".*

*Line 243: "Yicheng area is caused by Hg" instead of "Yicheng area is Hg contamination".*

*Lines 243-244: This is not true. According to table 2, As has a very similar $\Delta f(\alpha)$ value than that of Hg and the value for Zn is even greater than that of Hg.*

*Line 247: "because this element" instead of "as this element".*

*Line 252: Well, this is not exact. The element from Yicheng samples that showed the highest $\Delta f(\alpha)$ values was Zn, according to table 2.*

*Line 253: Remove "showing the distribution of Pb in the Daxing area and Hg in the Yicheng area" since it is already said in the former sentence.*

*Lines 255-279: This portion of text is a very poor discussion of your results. You did not discuss anything about Daxing contamination. It is also funny that you talk about Hg contamination in Yicheng but the concentrations of this element were greater in the samples of Daxing (see table 1). I am also not sure about the need of performing a multifractal analysis for obtaining these results; a simple geostatistical approach would be enough.*

*Lines 283-286: Please, re-phrase this caption, it is not clear.*

*Lines 289-291: I would change the caption of this figure to "Filled contour map obtained by inverse distance weighted interpolation showing the spatial distribution of soil Pb concentrations in the Daxing area".*

*Lines 294-296: I would modify the caption of this figure to "Filled contour map obtained by inverse distance weighted interpolation showing the spatial distribution of soil Hg concentrations in the Yicheng area".*

*Lines 299-302: I would change the caption of this figure to "Filled contour map obtained by inverse distance weighted interpolation showing the spatial distribution of soil Cu concentrations and the location of breeding facilities in the Yicheng area".*

*Conclusions:*

*Line 306: Include "the latter" after "although".*

*Line 307: Remove "for the soil geochemical data".*

*Line 309: Remove the word "value" before "changes".*

*Line 310: There is a mistake here; according to table 2, the Δf(α) in Yicheng decreased as Zn>Hg>As>Cd>Pb>Cu instead of Hg>Zn>As>Cd>Pb>Cu as is reported here.*

*Lines 314-319: However, the Hg concentrations in soils from the Daxing area were greater than in Yicheng.*

*Lines 320-326: I am not sure about this conclusion. Further explanations are needed in the discussion section to state this.*

*References:*

*I did not detect any missing references or citations in the text.*

*Line 343: Use the full name of the journal; in this case it should be "Computers and Geosciences" instead of "Comput. Geosci.".*

*Lines 346-347: It should be spelled with a capital letter: "University of Geosciences".*

*Line 353: It should be spelled with a capital letter: "Journal of Hazardous Materials".*

---

## Author Comment (AC1) · 26 May 2016

RC1: Anonymous Referee #1 This small note describes a good experimental dataset collected in two contracted landuse areas. The manuscript has to be amended with hypothetical explanation of processes causing the observed multifractal properties. Otherwise it is purely descriptive narrative that does add to science. The recommendations of soil reclamation have to be substantiated with suggestions of reclamation techniques and purposes of reclamation, otherwise these recommendations are superfluous. Note that references are mostly old, all older than 5 years.

We thank the reviewer for their suggestions. Our analysis indicates that different cleanup and remediation approaches are needed to resolve the issues relating to the differing heavy metal pollution in these areas, rather than a single approach to resolving heavy metal pollution. We have amended the text to reflect this as follows: "A significant amount of different remediation approaches can be used to resolve the issues of heavy metal soil contamination (e.g., Bech et al., 2014; Koptsik, 2014). The results presented in this study suggest that physical and chemical approaches (soil removal, soil vitrification, soil consolidation, electroremediation, soil washing) are more appropriate for the remediation of heavy metal contaminated soil in the Daxing area, especially in areas with significant heavy metal pollution, whereas the differing type of soil contamination in the Yicheng area could be more efficiently treated using microremediation and phytoremediation, primarily as the agriculture in this area requires a rapid reduction in the mobility and biological availability of heavy metals in the soils in this area (Mulligan et al., 2001; Wang et al., 2006)".

Please find the revised manuscript in the supplement.

Please also note the supplement to this comment:
http://www.nonlin-processes-geophys-discuss.net/npg-2016-15/npg-2016-15-AC1-supplement.pdf

**Supplement:**

[revised manuscript text omitted]

---

## Author Comment (AC2) · 26 May 2016

**GENERAL COMMENTS**

Interesting article that uses the multifractal spectrum to assess possible soil contamination by industrial and agricultural activity in two regions of China. An excellent data collection work and proper use of the chosen methods. The proposed use of the distribution of alpha singularity exponents to evaluate diffusion of contaminants in the soil is valid, but the conclusions require more robust criteria of causality. The main conclusions outlined by the authors are based on correlations and comparisons that were not carefully evaluated. The authors use visual and inaccurate comparisons to validate important statements in the paper argument. The following is a detailed description of

suggestions to improve the work.

SPECIFIC COMMENTS

Line 138: I would say it suggests a non-normal distribution a priori, only. The possible fractal/multifractal pattern is something to be evaluated a posteriori.

Thanks for the comments provided by J. Miranda. We have revised this sentence as follows: "All of the elements (barring Pb and Cu in the Yicheng area) in both the Yicheng and Daxing areas yield histograms that are positively skewed and contain some outliers (Fig. 2), indicating that these data have non-normal and potentially fractal- or multifractal-type distributions."

210: Why did you choose these values of q? Is there any argument (e.g., when the Dqxq curve stabilises)?

A range of q values between $-10$ and 10 with an interval of 1 is commonly used in these types of studies (Gonçalves et al., 2001; Xie et al., 2004; Dathe et al.,2006). To ensure that the results are reproducible we also replicated this analysis using a range of q values from $-1$ to 1 with an interval of 0.1, which yielded the same conclusions to the original range of q values.

Line 227: A comparison between the $\Delta f(\alpha)$ of the locations is considered here. The authors claim significant differences comparing only the order of the metals, sorted by $\Delta f(\alpha)$. Here a paired comparison statistic could prove the significant difference between areas.

We have added some text at lines Line 246-259 to compare the differences in heavy metal pollutions in the Daxing and Yicheng areas.

232,234: In my point of view, Figure 3 shows no sufficient evidence to conclude about correlations between the spectra of the two regions. A correlation test between $\Delta\alpha$ (left and right) in Daxing and $\Delta\alpha$ (left and right) in Yicheng could give more support to the argument.

We want to use Fig. 3 to show the differences shapes of the spectra between the two different parts of the study area as well as demonstrating the different multifractal characteristics of the heavy metal pollution in these areas. However, we have revised the manuscript using three parameters ($\Delta\alpha$, $\Delta$f(a) and $\tau$"(1)) to compare the heterogeneous patterns and degree of multifractality of the different heavy metals and areas, and we have rephrased the text to ensure this approach is described clearly.

255-257: A logistic correlation could substantiate the statement of significant correlation between the location of industrial/agricultural facilities and metals concentrations. 258-260: A very interesting hypothesis, associating the asymmetry of the spectra with the presence of anthropic actions. Whereas the single symmetric spectrum found was the Cu in the Yicheng area, we would expect a not significant logistics correlation between the presence of agricultural facilities and concentration of Cu in Yicheng (map in Figure 6) and significant correlation in the asymmetry cases. These tests would substantiate the argument of using multifractal for evaluation of anthropogenic changes.

We thank the reviewer for their suggestions, and we have added a new figure (Fig. 7) to show the relationship between heavy metal concentrations and the number of facilities in each area. This figure demonstrates a very good positive spatial correlation between the agricultural facilities in the Yicheng area and the high Hg concentration areas in this region and an even better positive spatial correlation between agricultural facilities in the Yicheng area and the high Cu concentration areas in this region. However, although this figure cannot show the degree of heavy metal pollution, it does demonstrate the spatial correlation between the location of industrial/agricultural facilities and areas of high metal concentrations, indicating a significant logistical correlation between the multifractality of the datasets and the industrial and agricultural activities in this area.

TECHNICAL COMMENTS

143: Just a suggestion: Make the legend a bit clear. The legend information is spread in the figure.

We have edited the legend in Fig. 1 as suggested.

197: I would say more heterogeneous patterns, given the non-binary feature of heterogeneity.

We changed the words "heterogeneous distribution patterns" to "more heterogeneous patterns".

214: "that describing the multifractality" - Unnecessary text.

We have removed "that describing the multifractality" from the table caption.

221: The f spectrum is only another way to characterise your set. I am not sure if 'best measure' is the most suitable term.

We have used three indexing methods $\Delta\alpha$, $\Delta f(a)$ and $\tau''(1))$ in the revised paper to allow a better analysis of the multifractal characteristics of the heavy metal pollution in soil in urban or developed areas.

234: Asymmetry concept could be better explained, it is presented in a way which might lead to misunderstandings. I would suggest an explanation based on the equations of the lines 191, 192 and 193.

We have added an additional comment in brackets as per the equation between lines 162-164 as follows: ($\Delta\alpha L$ is significantly different from $\Delta\alpha R$, equations 5-6).

238: Just two missed commas – "All of the heavy metals analyzed during this study, barring Hg, have higher $\Delta f(\alpha)$ values in soils from the Daxing area, with Hg having higher values in soils from the Yicheng area (Table 2)."

We have added the two missing commas.

241: "The only significant heavy metal pollution associated with the agricultural activity in the Yicheng area WOULD BE the Hg contamination".

We have revised this sentence according to the suggestions provided by both Miranda

and Miras Avalos as follows: "This suggests that the industrial activities in the Daxing area generate multi-element heavy metal soil contamination, whereas the only significant heavy metal pollution associated with the agricultural activity in the Yicheng area is Hg contamination."

Please find the revised manuscript in supplement.

Please also note the supplement to this comment:
http://www.nonlin-processes-geophys-discuss.net/npg-2016-15/npg-2016-15-AC2-supplement.pdf

---

## Author Comment (AC3) · 26 May 2016

Specific comments to the authors:

Please, organize the manuscript in Introduction, Materials and Methods, Results and Discussion, and Conclusions.

Title:

I suggest the authors to slightly modify the title of their manuscript to: "Comparison of the multifractal characteristics of heavy metals in soils within two areas of contrasting economic activities in China".

We thank the reviewer for their suggestion and have modified the title of our manuscript as follows: "Comparison of the multifractal characteristics of heavy metals in soils within two areas of contrasting economic activities in China".

Abstract:

The abstract is too long from my point of view.

We have shortened the abstract to highlight the main findings of our research.

Lines 16-17: "causing" instead of "that can have".

We have used "causing" instead of "that can have".

Line 23; "allows deeper interrogation", this expression is not clear. Please, re-phrase it.

We have revised this sentence to "Here, we present the results of a heavy metal (Cu, Pb, Zn, Cd, As and Hg) soil geochemical survey and use these data to evaluate and compare the characteristics of heavy metal pollution in soil in urban or developed areas."

Lines 26-28: "This study focuses...", this sentence can be removed since its information is reported in the next one.

We have removed this sentence.

Line 29: Include "(industrial)" after "Daxing" and "(agricultural)" after "Yicheng".

We have changed this sentence to "This study uses a multifractal spectral technique to identify the multifractality in the geochemistry of soils within the industrial Daxing and agricultural Yicheng areas of Anhui Province ".

Lines 31, 32 and 38: Use $\alpha$ instead of a in $\Delta f(\alpha)$, please.

We have used $\alpha$ instead of a in $\Delta f(\alpha)$.

Line 33: There is a mistake here; according to table 2, the $\Delta f(\alpha)$ in Yicheng decreased as Zn>Hg>As>Cd>Pb>Cu instead of Hg>Zn>As>Cd>Pb>Cu as is reported here.

We thank the reviewer for pointing out this mistake and for providing us with a new idea. Previously we thought that using f($\alpha$) to study contamination was sufficient, but we now realise that this is not sufficient and that $\Delta\alpha$, $\Delta f(\alpha)$ and $\tau$"(1) values also reflect different aspects of multifractality. As such, we have used $\Delta\alpha$, $\Delta f(\alpha)$ and $\tau$"(1) together in the revised manuscript to study and evaluate the multifractality of heavy metal contamination in the study area.

Line 34: I would remove the word "geochemical".

We have removed the word "geochemical".

Line 36: "clearly different" instead of "distinctly different".

We have changed "distinctly different" to "clearly different".

Lines 44-45: I would remove "rather than a single approach to heavy metal pollution" since it is not needed.

We have removed these words.

Introduction:

This section is not clear, the state-of-the-art is not put into context and thus the introduction seems out of focus. Moreover, this section begins with a list of references because in the first 6 lines, authors cited 13 references.

We have reorganized the introduction as recommended by the reviewer.

Line 51: I would remove "recently".

We have removed "recently".

Lines 58-59: "the factors controlling the distribution" instead of "the controls on the distribution".

We have changed "the controls on the distribution" to "the factors controlling the distribution".

Line 62: "in soil properties" instead of "in the characteristics of soils".

We have changed "in the characteristics of soils" to "in soil properties".

Line 63: Remove "and".

We have removed "and".

Lines 63-65: Please, check English, this sentence is unclear.

We have reorganized this sentence.

Lines 67-68: "but also in the analysis of" instead of "but can also be used in the analysis of".

We have changed "but can also be used in the analysis of" to "but also in the analysis of".

Line 70: "and thus" instead of "meaning that".

We have changed "meaning that" to "and thus".

Lines 73-75: Please, re-phrase this sentence. It is not clear what you mean and must be put in context with the former sentence.

We have reorganized this sentence.

Line 77: Please, define "C-A" when first used.

We have used "Concentration-Area" instead of "C-A".

Line 79: Please, define "S-A" when first used.

We have used the " Spectral density-Area" instead of "S-A".

Lines 78-83: This is not clear, please, revise it.

We have reorganized this sentence.

Line 86: Remove "provincial".

We have removed this word.

Line 87: Remove "areas".

We have removed this word.

Line 88: "activities" instead of "activity".

We have changed "activity" to "activities".

Lines 88-96: This portion of the text is repetitive and unclear. Please, revise it and state clearly the aims of your study.

We have reorganized these sentences to make the text more clear and more concise as follows:

"Here, we use multifractal techniques to determine the multifractal characteristics of the distribution of heavy metals in soils in these areas, using three multifractal parameters ($\Delta\alpha$, $\Delta f(\alpha)$ and $\tau''(1)$) to analyse and compare the degree and characteristics of the multifractality of heavy metal contamination in soils associated with the anthropogenic activities in this region. The results will further enable and inform future planning for any necessary remediation of these soils in the Daxing and Yicheng areas."

Study area and geochemical data:

Line 99: Include "it" before "has". Line 102: "industrial areas of Hefei" instead of "industrial bases of the Hefei area".

We have added "it" before "has" and used "industrial areas of Hefei" instead of "industrial bases of the Hefei area".

Line 103: Remove the word "industrial".

We have removed this word.

Line 105-106: "In contrast, the town of Yicheng focuses its economic activities on agricultural production" instead of "In contrast, the town of Yicheng is agricultural, with the economy of the town focused on agricultural production".

We have used "In contrast, the town of Yicheng focuses its economic activities on agricultural production" instead of "In contrast, the town of Yicheng is agricultural, with the economy of the town focused on agricultural production".

Line 107: "ornamentals" instead of "flower planting".

We have used "ornamentals" instead of "flower planting".

Line 110: I do not understand what you mean by "natural mineralization".

Here, we want to show that the soil in these areas are not influenced by mineralization or deposits.

Lines 110-111: "(< 20 cm depth)" instead of "(<20 cm below surface)".

We have used "(<20 cm depth)" instead of "(<20 cm below surface)".

Line 114: "was air-dried" instead of "was dried in air".

We have used "was air-dried" instead of "was dried in air".

Line 117: Remove "in the soil samples described above", remove also "during this study".

We have removed these words.

Line 119: "whereas Hg and As concentrations were determined" instead of ""with Hg and As concentrations determined".

We have used "whereas Hg and As concentrations were determined" instead of "with Hg and As concentrations determined".

Lines 122-125: You repeat too many times the word "analysis", sometimes you can use the synonym "determinations".

We thank the reviewer for this suggestion and have revised this sentence to "The accuracy of these data was monitored by repeat determinations of standards and replicate determinations of sub-sets of samples using instrumental neutron activation analysis (INAA). Analytical precision was monitored using determinations of variance of the results obtained from duplicate analyses."

Lines 116-125: Have you got references for the analytical methods? If so, please, add them to this portion of the text.

We think these analytical methods are well known as they have been used for a significant period of time and we do not want to make the reference list longer; as such, we have not made any specific reference to these techniques in the manuscript.

Line 126: "2.3. Results", this should be a section after the explanations of the materials and methods used.

We have moved this paragraph to Section 4.

Line 127: "A statistical summary" instead of "The results of a statistical analysis".

We have used "A statistical summary" instead of "The results of a statistical analysis".

Line 134: "the natural background". Maybe you should indicate what was the natural background.

Our original phrasing was not accurate; as such, we have changed this sentence to "This also suggests that samples from the Daxing area containing elevated concentrations of heavy metals were probably contaminated by anthropogenic activity."

Line 136: I would include Pb with Cu for the Yicheng area since the distribution of its concentrations in soils seems to follow a normal distribution as well.

We thank the reviewer for their suggestion and have revised the manuscript appropriately.

Line 137: I would include "(Fig. 2)" after "outliers".

We have moved "(Fig. 2)" after "outliers".

Line 138: I am not sure, I agree that they are non-normal but how can you tell from the histograms that they are fractal?

We are only speculating that these data have fractal distributions in this section; as such, we have changed the text to reflect this as follows: "indicating that these data have non-normal and potentially fractal- or multifractal-type distributions."

Line140: Remove "(Fig. 2)" from here.

We have removed "(Fig. 2)" from here.

Lines 143-146: I would rephrase this figure caption to "Location of Hefei in central-eastern China (a); location of the study areas within Hefei (b); 1 x 1 km grid for soil sampling in the towns of Daxin (c) and Yicheng (d)".

We have changed this figure caption.

Line 148: Re-phrase the title of this table to "Summary statistics of soil heavy metal concentrations from the Daxing and Yicheng samples".

We have changed this figure caption.

Table 1: Skewness and kurtosis are not concentrations and they are dimensionless. I would put the units below each column title, I mean below "minimum", "maximum", "mean" and "standard deviation". I would remove "Concentrations" from the table.

We thank the reviewer for this suggestion and have revised this table accordingly.

Mutifractal spectrum analysis:

Equations should be numbered.

We have numbered all of the equations in the text.

Line 159: "the factors controlling the distribution" instead of "the controls on the distribution". What do you mean by "of key elements within data".

We have changed the "the controls on the distribution" to "the factors controlling the distribution".

The key elements we want to express are the important study objects within the data, such as the heavy metals, nutrition component, porosity of soil, and so on.

Line 160: Remove the word "multifractal". "$f(\alpha)$" instead of "f(a)".

We have remove the word "multifractal" and now use "$f(\alpha)$" instead of "f(a)".

Line 163: Remove "of estimating f(a) values" since it is not needed.

We have remove the words "of estimating f(a) values".

Line 170: "different from 0" instead of "that $\neq 0$".

We have changed to this to "different from 0" instead of "that $\neq 0$" as suggested by the reviewer.

Lines 173-174: Move "within a dataset" to after "statistical estimation".

We have moved "within a dataset" to after "statistical estimation"

Line 183: "different from 0" instead of "that $\neq 0$".

We have changed this to "different from 0" from "that $\neq 0$"

Lines 184-185: Use alpha ($\alpha$) instead of a when referring to the multifractal spectra.

We have changed this throughout the manuscript.

Line 194: "spectrum is" instead of "spectrum are".

We have used "spectrum is" instead of "spectrum are".

Line 197: Use alpha ($\alpha$) instead of a when referring to the multifractal spectra.

We have changed this throughout the manuscript.

Line 201: "by the following" instead of "using the following".

We have used "by the following" instead of "using the following".

Calculation processes and discussion:

Lines 205-209: This has already been said in the former section.

We have removed these sentences from this section.

Line 214: Remove "that" and "all of".

We have removed these words.

Lines 217-223: This description should be greatly improved. Check English, please. Only Cu and Pb for Yicheng area have $\tau$"(1) values lower than -0.01.

We have revised this sentence as follows as per the reviewer's comments:

"The multifractal analytical results shown in Table 2 indicate that all of the elements (barring Cu in the Yicheng area) are characterized by a wide range of $\alpha$ values (i.e. have high $\Delta$ïĄ̨ values), have $\tau$"(1) values less than –0.01 (barring Cu and Pb in the Yicheng area) and have $\Delta f(\alpha)$ values larger than 0.5 (barring Cu in the Yicheng area), all of which indicate that these elements have highly multifractality within the soils in these two areas".

Line 225: Use "indices" instead of "elements". You are not talking about the elements but the multifractal indices that you obtained.

The revised version of this manuscript uses three multifractal parameters to study the multifractality of the heavy metal distribution in soils in the study area. We have amended the text to reflect this as follows:

"The overall amount of multifractality within the soil geochemical data for the Daxing area decreases as follows: Pb>Cd>As>Zn>Hg>Cu, whereas the overall amount of multifractality within the soil geochemical data for the Yicheng area decreases as follows: Hg>Zn>As>Cd>Pb>Cu".

Line 226: "decrease" instead of "decreases". There is a mistake here, Zn have a greater f(ïĄą) value than Hg for the Yicheng samples.

We have corrected the text and have discussed all three of the multifractal parameters within the text.

Line 227: This has already been observed in the statistical summary.

We have removed this sentence. However, we have also compared the differences between the statistical summary and the results of our multifractal analysis as follows: "Table 3 indicates that the Zn data has largest standard deviation and a moderate coefficient of variation within the Daxing area, but the $\Delta\alpha$ and $\Delta f(\alpha)$ values for these Zn data indicate only weak multifractality compared with other heavy metals. In comparison, the Hg data for soils in the Yicheng area yields the lowest standard deviation but the largest $\Delta\alpha$ and $\tau"(1)$ values, indicating these Hg data have strong multifractality. These differences indicate that the multifractal parameters $\Delta\alpha$, $\Delta f(\alpha)$ and $\tau"(1)$ reveal new information about the nonlinear variability and the characteristics of these geochemical data compared to the analyses afforded by classic basic statistics".

Line 229-231: In fact, you are plotting these data.

We have deleted this sentence to make the text more logical.

Lines 232-234: This is not clear. Please, re-phrase it.

We have revised this paragraph, as follows:

"Multifractal spectra combine the singularity exponent $\alpha$ and the corresponding fractal dimension f($\alpha$) to generate a multifractal spectrum with an inverse bell shape (Fig. 3). All of these multifractal spectra are also asymmetric ($\triangle\alpha$L is significantly different from $\triangle\alpha$R, equations 5-6) (barring the Cu data for soils from the Yicheng area), indicating that the soils containing low and high concentrations of these elements are not evenly distributed within the study area (as is expected for areas containing point source pollutants like factories or animal breeding facilities)."

Line 235: Remove "for all elements".

We have removed "for all elements".

Lines 235-240: I am not sure about understanding this. Please, re-phrase it.

We have revised this sentence as follows:

"All of these multifractal spectra are also asymmetric ($\triangle\alpha$L is significantly different from $\triangle\alpha$R, equations 5-6) (barring the Cu data for soils from the Yicheng area), indicating that the soils containing low and high concentrations of these elements are not evenly distributed within the study area (as is expected for areas containing point source pollutants like factories or animal breeding facilities)."

Line 241: "heavy metal contamination of soil" instead of "heavy metal contamination soil contamination".

We have used "heavy metal contamination of soil" instead of "heavy metal contamination soil contamination".

Line 243: "Yicheng area is caused by Hg" instead of "Yicheng area is Hg contamination".

We have used "would be mainly caused by Hg" instead of "is Hg contamination".

Lines 243-244: This is not true. According to table 2, As has a very similar $\Delta$f($\alpha$) value than that of Hg and the value for Zn is even greater than that of Hg.

We have updated this and now use three multifractal parameters to discuss the results of our study.

Line 247: "because this element" instead of "as this element".

We have used "because this element" instead of "as this element".

Line 252: Well, this is not exact. The element from Yicheng samples that showed the highest $\Delta f(\alpha)$ values was Zn, according to table 2.

We have updated this and now use three multifractal parameters together to discuss the results of our study.

Line 253: Remove "showing the distribution of Pb in the Daxing area and Hg in the Yicheng area" since it is already said in the former sentence.

We have removed "showing the distribution of Pb in the Daxing area and Hg in the Yicheng area" from the sentence.

Lines 255-279: This portion of text is a very poor discussion of your results. You did not discuss anything about Daxing contamination. It is also funny that you talk about Hg contamination in Yicheng but the concentrations of this element were greater in the samples of Daxing (see table 1). I am also not sure about the need of performing a multifractal analysis for obtaining these results; a simple geostatistical approach would be enough.

We have added Table 3 and Fig. 5 as well as associated text to enhance our discussion of our results, including comparing the differences between the results of purely statistical summaries and multifractal analysis. Our study indicates that multifractal modeling and the associated generation of multifractal parameters is a useful approach for the evaluation of heavy metal pollution in soils and the identification of major sources of heavy metal contamination.

Lines 283-286: Please, re-phrase this caption, it is not clear.

We have rephrased and simplified this caption to make it more clear.

Lines 289-291: I would change the caption of this figure to "Filled contour map obtained by inverse distance weighted interpolation showing the spatial distribution of soil Pb concentrations in the Daxing area".

We thank the reviewer for their suggestion and have amended the caption for Fig. 4 appropriately.

Lines 294-296: I would modify the caption of this figure to "Filled contour map obtained by inverse distance weighted interpolation showing the spatial distribution of soil Hg concentrations in the Yicheng area".

We have changed the caption of Fig. 5.

Lines 299-302: I would change the caption of this figure to "Filled contour map obtained by inverse distance weighted interpolation showing the spatial distribution of soil Cu concentrations and the location of breeding facilities in the Yicheng area".

We have changed the caption of Fig. 6.

Conclusions:

Line 306: Include "the latter" after "although".

We have included "the latter" after "although".

Line 307: Remove "for the soil geochemical data".

We have removed these words.

Line 309: Remove the word "value" before "changes".

We have removed the word "value" before "ranges".

Line 310: There is a mistake here; according to table 2, the $\Delta f(\alpha)$ in Yicheng decreased as Zn>Hg>As>Cd>Pb>Cu instead of Hg>Zn>As>Cd>Pb>Cu as is reported here.

We have updated the conclusions to include this.

Lines 314-319: However, the Hg concentrations in soils from the Daxing area were greater than in Yicheng.

We have updated the conclusions to include this.

Lines 320-326: I am not sure about this conclusion. Further explanations are needed in the discussion section to state this.

We have rewritten the conclusions to make them more clear and to reflect our enlarged discussion section.

---

## Referee Report (RR1)

The revised version of the manuscript with reference NPG-2016-15-R1 and entitled "Comparison of the multifractal characteristics of heavy metals in soils within two areas of contrasting economic activities in China" authored by X. Li, F. Yuan, S.M. Jowitt, X. Li, T. Zhou, J. Zhou, X. Hu, and Y. Li and submitted to the Special Issue "Multifractal analysis in soil systems" to be published in Nonlinear Processes in Geophysics represents an improvement from the former version submitted to the journal. Authors have addressed the comments and suggestions made by the reviewers and English has been carefully revised.

However, there are still several issues to be corrected prior to an eventual publication in the journal. I detected a mistake in the results section and a number of them within the methodology, besides the discussion is weak and uses wrong concepts that authors must correct in order to get reliable conclusions. The last figure does not reflect what the authors imply in their discussion and should be re-arranged. Apart from this, a number of minor issues have been raised in the introduction section.

Therefore, I still advice for a major revision prior to the acceptance of the manuscript.

In the following pages, I provide the authors with a number of suggestions/comments for improving their manuscript.

**Specific comments to the authors:**

*Abstract:*

*The abstract is rather long and redundant from my point of view.*

*Line 23; "within urban or developed areas", is this really needed?*

*Line 24: I would remove "an area" since it is not needed.*

*Line 26: "geochemistry of soils", this is to wide and not really what you have done since you only applied multifractal formalisms to heavy metal concentrations.*

*Lines 27-33: I think this portion could be re-phrased and reduced. Besides, there is a mistake in line 32, according to table 3, the overal multifractality in Yicheng decreased as Hg>As>Zn>Cd>Pb>Cu and not as Hg>Zn>As>Cd>Pb>Cu as is reported here. In line 33, it should be "indicate" instead of "indicates" since you refer to "differences".*

*Lines 38-40: I would remove this sentence: "The larger values […] to industrial activities than agriculture".*

*Line 46: I do not agree with this statement. You did not identify any source of pollution using multifractality, the spatial analysis was performed using a deterministic method (inverse distance weigthing).*

*Introduction:*

*I did not like this way of introducing because you did not set the basis for performing your study. Anyway, this a personnal opinion. However, certain objective remarks need to be addressed:*

*Line 54: I would remove "in recent years". You cited here works from 1997, that is 20 years ago, therefore, I do not think is very recent. By the way, in what order did you put your citations? It is neither chronological nor alphabetical! Please, edit according to journal's instructions.*

*Lines 55-59: Please, try to reduce this sentence because it is too long.*

*Lines 61-62: I would remove "using multifractal techniques to" and substitute "determine" for "determining".*

*Line 63: I would use other term instead of "further". Maybe "enhance" or "improve".*

*Line 64: What activities are you referring to?*

*Lines 65-74: This is somewhat confusing since you did not define any threshold, separated anomalies or identified behaviours in your study. At least I did not understand it that way.*

*Line 76: Remove "and so on".*

*Line 78: "polluntation"??? Do you mean "pollution"?*

*Line 80: Remove "be used".*

*Lines 74-84: What order did you follow for citations? It is neither chronological nor alphabetical.*

*Lines 87-90: In this sentence the word "multifractal" is repeated four times, please, re-phrase.*

*Line 91: Remove "the" before "anthropogenic".*

*Study area and geochemical data:*

*Line 114: Remove "during this study, with"and use a dot to separate the sentence after "determined". Then begin the new sentence as "The concentrations of".*

*Line 115: Remove "concentrations" and use "were" before "determined".*

*Line 116: Use "were" before "determined".*

*Lines 114-117: Please, add references for the methods used for heavy metal determination.*

*Lines 119-122: This is not clear, please, re-phrase it.*

*Multifractal spectrum analysis*

*Lines 148-149: Again, order of citations!*

*Line 151: Since you used the gliding box method, why explaining the calculation of the box-counting method in lines 140-144?*

*Line 154: Please, improve the readability of this equation.*

*Line 159: "f(α)" does not appear in equation 4.*

*Lines 160-161: What is "q" in these equations?*

*Lines 162-163: These symbols do not appear in the equations 3 and 4, why beginning with "where"? You must specify the meaning of the symbols in each equation, otherwise, readers will not know what are you describing mathematically.*

*Lines 173-174: "multifractality associated with ordinary spatial analysis parameters", what parameters? What is the relation?*

*Line 178: "is the box-counting dimension", but you were using the gliding-box approach. I am lost.*

*Line 179: "smaller values". Not clear, smaller than what? Positive or negative?*

*Line 181: "used" instead of "use".*

*Line 182: "heterogeneous patterns", of what?*

*Lines 183-184: "as well as enabling the comparison of the distribution of differing elements in the soils in this region". If you say so, but I am not so sure, in fact, you performed this using inverse distance weighting interpolation.*

*Geochemical analysis results*

*Line 187: You do not indicate that means were also higher for the Daxing area. Besides, they are also higher in this area for Hg.*

*Line 196: "yielded concentration histograms" instead of "yield histograms".*

*Lines 198-200: And also that these concentrations depended on the type of human activities developed within each area.*

*Calculation processes of multifractal spectrum and discussion*

*Lines 210-211: This is already explained in the description of the multifractal analysis that has been carried out.*

*Lines 211-212: "used a range of q values from –10 to 10", did you select this range? Besides, you did not explain what "q" is.*

*Line 216: I would remove "showing the multifratal characteristics of all".*

*Line 217: I would remove "(barring Cu)"and add "Daxing" to the figure caption.*

*Lines 219-220: I would remove "combine the singularity exponent α and the corresponding fractal*

*dimension f(α) to generate a multifractal spectrum with" and use just "showed".*

*Line 221: "are also" should be substituted for "were" and "is" for "was".*

*Line 223: "samples" instead of "the soils".*

*Line 226: Remove "analytical" and use "indicated" instead of "indicate".*

*Line 227: "were characterized" instead of "are characterized".*

*Lines 227-230: Please, re-phrase. Use the past tense and remove innecessary words.*

*Line 232: "had" instead of "have".*

*Lines 232-233: Are these differences significant?*

*Lines 235-236: "the significant heavy metal pollution associated with agriculture"; however, concentrations of Hg were greater in the industrial area.*

*Lines 237-240: I do not agree with this explanation. The high values for the multifractal indices used in this study just mean that in your data series high concentration values were very different from low concentrations for a given element.*

*Lines 241-243: I do not see why you only concentrate on Hg pollution in your discussion, what about the other elements?*

*Lines 243-244: "deleterious effects", on what?*

*Line 246: I would remove "within the soil samples".*

*Lines 249-251: This should be explained in the materials and methods. Besides, you should indicate that data were sorted within each area and not on both at the same time.*

*Line 260: I would remove "the analyses afforded by classic".*

*Lines 263-265: This is not clear, please, re-phrase it.*

*Line 267: I would change "parameters and coefficient of variation values" to "and basic statistic indices".*

*Table 3: Using standard deviation is of not very much use here since its effect would depend on the magnitude of your data. For instance Zn has a very high standard deviation compared to Hg, so logically, this index would give Zn always the first order. I suggest only using coefficient of variation, since this index is normalized for all elements.*

*Line 273: According to table 3, it should be "Hg>As>Zn>Cd>Pb>Cu".*

*Line 280: Why not performing this inverse distance weighting interpolation for the rest of the elements? Besides, be careful since here you are not using multifractals; however, your discussion is oriented as if this technique was multifractal.*

*Lines 284-285: Not exactly, only in the area where is a bunch of breeding facilities. In the case of*

*Pb, this concentration would depend on the type of industry involved.*

*Lines 287-290: What are they using as Hg source? It must appear from somewhere!!!*

*Lines 293-295: Please, re-phrase. This is not clear.*

*Line 296: "a significant" instead of "an significant".*

*Lines 297-299: I do not see this from your figure. In fact, the evolution of both elements is very similar at lower classes.*

*Lines 301-302: I do not totally agree. You can see that the shape of the curve is similar for Cu and Hg, only different for the greater classes.*

*Line 304: "can efficiently reflect the multifractality", of course, they are designed to do this.*

*Line 305: Remove "by".*

*Lines 317-319: This is unclear. Please, re-phrase it.*

*Line 320: Remove "especially in areas with significant heavy metal pollution". From my viewpoint, it is not needed.*

*Line 324: Remove "in this area".*

*Figure 4: In the upper left-hand side of the map there are some industries and the Pb concentration is rather low. Similar values are observed in the lower left-hand side of the map. This may indicate that Pb concentrations in soils depend more on the type of industry than on the fact that there is an industry, as you imply in your discussion.*

*Figure 5: In contrast with the former figure, in the right-hand side of this map, we can observe high concentrations of Hg in the soil, but there are no breeding facilities on this part of the map... Then, why do these high Hg concentrations appear? According to your discussion, there is a direct relationship between the existence of a breeding facility and the high concentrations of Hg observed.*

*Figure 6: Coinciding with the former map, there are very high Cu concentrations in the right-hand side of this map, why? What is over there?*

*Figure 7: To me, this graph is difficult to interprete. Both lines show correlations between number of facilities and Hg or Cu concentrations in soils. Besides, the caption is not clear. You talk about an anti-correlation and this is not observed in the graph.*

*Conclusions:*

*Line 355: According to table 3, the overal multifractality in Yicheng decreased as Hg>As>Zn>Cd>Pb>Cu and not as Hg>Zn>As>Cd>Pb>Cu as is reported here.*

*Lines 356-364: What about other problems caused by the other heavy metals? People in Daxing are inmune to heavy metal pollution?*

*Line 365: "The initial results", I do not understand why you termed your results as "initial".*

*Lines 366-367: "multifractal parameters can efficiently reflect the multifractality caused by industrial and agricultural activities", well, of course, multifractal indices are designed to do this.*

*Lines 369-370: "and the identification of major sources of heavy metal contamination". I do not agree, the identification of these sources was made through inverse distance weigthing interpolation, which is not a multifractal technique.*

---

## Editor Decision (ED1)

The revised version of the manuscript with reference NPG-2016-15-R2 and entitled "Comparison of the multifractal characteristics of heavy metals in soils within two areas of contrasting economic activities in China" authored by X. Li, F. Yuan, S.M. Jowitt, X. Li, T. Zhou, J. Zhou, X. Hu, and Y. Li and submitted to the Special Issue "Multifractal analysis in soil systems" to be published in Nonlinear Processes in Geophysics represents an improvement from the former version submitted to the journal. Authors have addressed the comments and suggestions made by the reviewers.

However, there are still several issues to be corrected prior to an eventual publication in the journal. I detected several mistakes in the reference list as well as some problems with English.

Therefore, I still advice for a minor revision prior to the acceptance of the manuscript.

In the following pages, I provide the authors with a number of suggestions/comments for improving their manuscript.

**Specific comments to the authors:**

*Abstract:*

*Line 22: "city. We used a multifractal" instead of "city and use a multifractal".*

*Line 25: Remove "for these soil geochemical data".*

*Lines 28-31: Please, consider re-phrasing this sentence. For instance, I propose to reduce it to "The degree of multifractality suggests that the differing economic activities in Daxing and Yicheng generate very different heavy metal pollution loads".*

*Introduction and overview of the study area:*

*Line 47: "may allow for assessing" instead of "can investigate".*

*Line 48: "with pollutants, as well as" instead of "with pollutants and as well as".*

*Line 61: Remove "in this case".*

*Lines 72-73: Please, check the order of citations here. Besides, please, consider reducing the number of citations seems I think there are far too many than necessary.*

*Study area and geochemical data:*

*Line 109: Remove "was" from "was monitored".*

*Multifractal spectrum analysis:*

*Check the readability of equation 1. Parenthesis and symbols are overlapping.*

*Lines 141-142: You should indicate what are $\alpha$ and $f(\alpha)$.*

*Line 146: "a given dataset" instead of "the dataset in question". Indicate the symbols that you used for "left and right branches".*

*Lines 148-150: Check the readability of equations 4 to 6. The "min" is not visible.*

*Lines 156-160: Re-phrase, not clear.*

*Line 164: "monofractal" instead of "single fractal".*

*Lines 165-168: Consider removing, already stated in lines 77-80.*

*Geochemical analysis results*

*Lines 173-174: Ok, but the minimum and the mean were higher in the Daxing area.*

*Line 185: I do not think that you need multifractals to discriminate this.*

*Calculation processes of multifractal spectrum and discussion*

*Lines 196-197: "using a range of q values from -10 to 10 with and interval of 1" should be stated in the "multifractal spectrum analysis" section.*

*Line 212: "have a high multifractality" instead of "have highly multifractality".*

*Line 215: It should be $\Delta\alpha$ instead of $\alpha$.*

*Line 222: "but not maximum", if you look at your histogram (figure 2) you will see that most of the samples from Yicheng were below 0.31 mg/kg of Hg, whereas in Daxing there are more samples above 0.5 mg/kg of Hg.*

*Line 229: What do you mean by "deleterious effects such as the heavy metal pollution of people, crops and animals"?*

*Line 236: "the elements within the samples from" instead of "different elements within".*

*Lines 238-245: Please, remove all references to standard deviation because it is not used for sorting the elements according to table 3.*

*Line 262: "indicate" instead of "indicates".*

*Line 276: "polluting more than others" instead of "more polluting than others".*

*Lines 296-297: How was spatial density assessed? It is not explained anywhere.*

*Lines 302-305: When you are talking about correlation value, do you refer to coefficient of correlation? Could you indicate the p-values for these correlations, please?*

*Line 308: "do" instead of "does".*

*Lines 308-311: Please, re-phrase, not clear.*

*Line 339: Include a parenthesis after "water".*

*Line 340: Remove "or other source".*

*Line 346: Inverse distance weighted interpolation is not mentioned or explained in the methodology.*

*Line 360: How was this density map created? This should be explained in the methods.*

*Table 4: How was this correlation matrix made? It is not explained in the methodology.*

*Lines 365-367: There is no figure 8.*

*Lines 368-371: Is this needed?*

*Conclusions:*

*Lines 374-376: Consider removing the first sentence.*

*Line 382: Remove "the overall order in soils".*

*Line 390: Remove the word "well".*

*Lines 393-396: I think that this sentence can be removed.*

*References:*

*Lines 417-418: Caniego et al. 2005 are not cited in the text. Please, remove.*

*Lines 429-431: Dathe et al. 2006 are not cited in the text. Please, remove.*

*Lines 445-447: Hasley et al. 1986 are not cited in the text. Please, remove.*

*Lines 469-474: McGrath et al. 2004 should come before Mulligan et al. 2001.*

---

## Author Response (AR2)

Dear Prof. Schertzer and Dr. Miras-Avalos,

We thank anonymous referee #1 and Jose Miranda for their careful and constructive reviews of our manuscript. We have uploaded our response as a supplement to the comments and have incorporated these changes to our revised manuscript. For clarity, we have used a blue font for the reviewer's text, a black font for our text, and italics for text that is included in the revised manuscript. We hope that after these revisions our manuscript will be considered suitable for publication in Nonlinear Processes in Geophysics.

**RC1: Anonymous Referee #1**

**Introduction:**
**I did not like this way of introducing because you did not set the basis for performing your study. Anyway, this a personnal opinion. However, certain objective remarks need to be addressed:**
**Line 54: I would remove "in recent years". You cited here works from 1997, that is 20 years ago, therefore, I do not think is very recent. By the way, in what order did you put your citations? It is neither chronological nor alphabetical! Please, edit according to journal's instructions.**
We have removed "in recent years" and have listed all of the citations in chronological order.

**Lines 55-59: Please, try to reduce this sentence because it is too long.**
We have reorganized this sentence and divided it to two sentences to make it clearer.

**Lines 61-62: I would remove "using multifractal techniques to" and substitute "determine" for "determining".**
We have removed "using multifractal techniques to" and substituted "determine" for "determining".

**Line 63: I would use other term instead of "further". Maybe "enhance" or "improve".**
We have used "improve" instead of "further".

**Line 64: What activities are you referring to?**
We have used "anthropogenic" to restrict the "activities".

**Lines 65-74: This is somewhat confusing since you did not define any threshold, separated anomalies or identified behaviours in your study. At least I did not understand it that way.**
This section introduces the potential uses of several different multifractal techniques and indicate that multifractal techniques are useful techniques for understanding the distribution of heavy metals in soils. As such, we state that "Here, we use multifractal spectra techniques and three parameters ($\Delta\alpha$, $\Delta f(\alpha)$ and $\tau''(1)$) to analyze and compare the degree and characteristics of the multifractality of heavy metal contamination in soils associated with anthropogenic activities in this region".

**Line 76: Remove "and so on".**

We have removed "and so on".

**Line 78: "polluntation"??? Do you mean "pollution"?**

This was a spelling error and we have corrected this.

**Line 80: Remove "be used".**

We have removed "be used".

**Lines 74-84: What order did you follow for citations? It is neither chronological nor alphabetical.**

We have put all of our citations in chronological order.

**Lines 87-90: In this sentence the word "multifractal" is repeated four times, please, re-phrase.**

We have rephrased this sentence.

**Line 91: Remove "the" before "anthropogenic".**

We have removed "the" before "anthropogenic".

**Study area and geochemical data:**

**Line 114: Remove "during this study, with"and use a dot to separate the sentence after "determined".**

**Then begin the new sentence as "The concentrations of".**

We have revised these sentences as suggested by the reviewer.

**Line 115: Remove "concentrations" and use "were" before "determined".**

We have removed "concentrations" and have included "were" before "determined".

**Line 116: Use "were" before "determined".**

We have used "were" before "determined".

**Lines 114-117: Please, add references for the methods used for heavy metal determination.**

We have added references to provide information on the methods used.

**Lines 119-122: This is not clear, please, re-phrase it.**

We have rephrased these sentences.

**Multifractal spectrum analysis**

**Lines 148-149: Again, order of citations!**

We have put all of our citations in chronological order.

**Line 151: Since you used the gliding box method, why explaining the calculation of the boxcounting method in lines 140-144?**

The boxcounting method is the basis of the gliding box method used during this study. As such, we originally explained this method as well as the gliding box method; however, in order to more clearly explain the methods used we have removed this paragraph as suggested by the reviewer and have renumbered our equations appropriately.

**Line 154: Please, improve the readability of this equation.**

We have updated this equation.

**Line 159: "$f(\alpha)$" does not appear in equation 4.**

We have revised this equation.

**Lines 160-161: What is "$q$" in these equations?**

$q$ is the order moment of the measure $\mu_i(\varepsilon)$. We have this description to in Line 134.

**Lines 162-163: These symbols do not appear in the equations 3 and 4, why beginning with "where"? You must specify the meaning of the symbols in each equation, otherwise, readers will not know what are you describing mathematically.**

We have deleted the "where" in the begining of this sentence.

**Lines 173-174: "multifractality associated with ordinary spatial analysis parameters", what parameters? What is the relation?**

We have changed this sentence to " In addition, local multifractality $\tau''(1)$, can also be used as a measure to quantitatively characterize the multifractality of a dataset using the equation 8, where ordinary spatial analysis functions (autocorrelation and semivariogram) are related to the low order statistical moments (0 to 2nd) that may determine $\tau''(1)$ (Cheng, 2006)".

**Line 178: "is the box-counting dimension", but you were using the gliding-box approach. I am lost.**

The gliding box method is derived from the box-counting method, meaning that the D involved in the box-counting box method is as same that used in the gliding-box approach.  The fact that we have removed the equation in question from the revised manuscript means that we have also changed this text from "the box-counting dimension" to "the gliding-box dimension".

**Line 179: "smaller values". Not clear, smaller than what? Positive or negative?**

The fact that $-D < \tau''(1) < 0$ means that the value of $\tau''(1)$ will be negative. As such, we have changed changed "smaller values" to "more negative values" to clarify this.

**Line 181: "used" instead of "use".**

We have used "used" instead of "use".

**Line 182: "heterogeneous patterns", of what?**

Heterogeneous patterns are ordered, complex, clustered patterns; to clarify this we have added. We have added this to the manuscript.

**Lines 183-184: "as well as enabling the comparison of the distribution of differing elements in the soils in this region". If you say so, but I am not so sure, in fact, you performed this using inverse distance weighting interpolation.**

We have changed this sentence to "as well as enabling the comparison of the multifractality of differing elements in the soils in this region".

**Geochemical analysis results**

**Line 187: You do not indicate that means were also higher for the Daxing area. Besides, they are also higher in this area for Hg.**

We have added "mean" to this line.

**Line 196: "yielded concentration histograms" instead of "yield histograms".**

We have used "yielded concentration histograms" instead of "yield histograms".

**Lines 198-200: And also that these concentrations depended on the type of human activities developed within each area.**

We have changed this sentence to "This means that multifractal techniques are highly suited for the characterization of the geochemistry of the soils discrimination of the differing types of human activities ongoing in each area.".

**Calculation processes of multifractal spectrum and discussion**

**Lines 210-211: This is already explained in the description of the multifractal analysis that has been carried out.**

We have rewritten lines 210-212 to clarify this

**Lines 211-212: "used a range of $q$ values from –10 to 10", did you select this range? Besides, you did not explain what "$q$" is.**

We did select $q$ values from –10 to 10 and have clarified this in the text. As described above, $q$ is the order moment of the measure, as described in Line 134. We have also rewritten lines 210-212.

**Line 216: I would remove "showing the multifratal characteristics of all".**

We have removed "showing the multifractal characteristics of all".

**Line 217: I would remove "(barring Cu)"and add "Daxing" to the figure caption.**

We have removed "(barring Cu)"and added "Daxing" to the figure caption.

**Lines 219-220: I would remove "combine the singularity exponent and the corresponding fractal dimension $f(\alpha)$ to generate a multifractal spectrum with" and use just "showed".**

We have used "showed" to make this sentence clearer.

**Line 221: "are also" should be substituted for "were" and "is" for "was".**

We have used "were" instead of "are also" and "was" instead of "is".

**Line 223: "samples" instead of "the soils".**

We have used "samples" instead of "the soils"

**Line 226: Remove "analytical" and use "indicated" instead of "indicate".**

We have removed "analytical" and used "indicated" instead of "indicate".

**Line 227: "were characterized" instead of "are characterized".**

We have used "were characterized" instead of "are characterized".

**Lines 227-230: Please, re-phrase. Use the past tense and remove innecessary words.**

We have re-phrased this sentence.

**Line 232: "had" instead of "have".**

We have used "had" instead of "have".

**Lines 232-233: Are these differences significant?**

We think these differences are significant; the Daxing area has $\Delta\alpha$, $\Delta f(\alpha)$ and $\tau''(1)$ values that are double the values for the Yicheng area for the majority of the elements analysed during this study.

**Lines 235-236: "the significant heavy metal pollution associated with agriculture"; however, concentrations of Hg were greater in the industrial area.**

We thank the reviewer for pointing this out. As such, we have added the following section to the manuscript: "Although the mean concentrations of Hg in soils are greater in Daxing area, all of the multifractal parameters determined during this study ($\Delta\alpha$, $\Delta f(\alpha)$ and $\tau''(1)$) indicate that the Hg data in the Daxing area has a lower multifracticality than the Hg data in the Yicheng area. The Yicheng area is heavily agricultural, meaning that the agricultural activities in this area may be both concentrating Hg as well as contaminating soils. In addition, although the concentrations of Hg in the Yicheng area are lower than in the soils in the Daxing area, both are of significance. Indeed, the lower concentrations in the Yicheng area may be of more concern than the higher concentrations in the Daxing area, as the agricultural activity in this area may lead to greater human intake of Hg than from the soils in the mainly industrial Daxing area, a factor that could lead to serious health issues (e.g. Minamata disease) caused by the potential concentration of Hg up the food chain. This indicates that soils in both areas may well require control and remediation."

**Lines 237-240: I do not agree with this explanation. The high values for the multifractal indices used in this study just mean that in your data series high concentration values were very different from low concentrations for a given element.**

We thank the reviewer for their suggestion and have made our discussion more rigorous, as evidenced by the revision of this sentence as follows: " The $\Delta f(\alpha)$ and $\alpha$ values of Hg in the Yicheng area are larger than the values for all other elements in this area as well as some of the elements in the Daxing area, indicating both the prevalence and significant degree of agricultural Hg contamination in the Yicheng area, even considering the lower overall (but not maximum) concentrations of Hg within the Yicheng area compared to the Daxing area. This contamination should be considered a priority in terms of remediation, because the interaction between the agricultural activity in the Yicheng area and this Hg pollution could seriously impact human health, as Hg is preferentially concentrated upward in the food chain (e.g. (Jiang et al., 2006)). This means that although contamination in both areas needs to be evaluated further and should be remediated to avoid any deleterious effects such as the heavy metal pollution of people, crops and animals, the fact that the Hg contamination in the Yicheng area may be more bioavailable and may have a larger effect on the population of this region (as a result of the agricultural activity in this area) means it should be considered a priority".

**Lines 241-243: I do not see why you only concentrate on Hg pollution in your discussion, what about the other elements?**

The main aim of this paper is to explain that the three multifractal indices used during this study are useful tools for the evaluation of the degree of influence on the heavy metals in soils caused by human activities. We focus on Hg in the Yicheng area rather than the other elements as Hg has higher $\Delta f(\alpha)$ and $\Delta \alpha$ and lower $\tau''(1)$ values than all of other elements in the Yicheng area and some of the elements in the Daxing area, even though the mean Hg concentrations in the Yicheng area are lower than in the Daxing area. These characteristics mean we have focused on the Hg contamination of the soils in the Yicheng area.

**Lines 243-244: "deleterious effects", on what?**

We mean the heavy metal pollution of people, crops and animals, and have stated this in the text.

**Line 246: I would remove "within the soil samples".**

We have removed "within the soil samples" in the title of Table 2.

**Lines 249-251: This should be explained in the materials and methods. Besides, you should indicate that data were sorted within each area and not on both at the same time.**

The standard of the order is described in lines 151 and 160 and we have revised some of this text to make it clearer. We have also changed this sentence to "In order to compare variations in multifractality, different elements within Daxing and Yicheng area were sorted by $\Delta\alpha$, $\Delta f(\alpha)$ and $\tau''(1)$ parameters respectively, in addition to sorting by basic statistics such as standard deviation and coefficient of variation values (Table 3)".

**Line 260: I would remove "the analyses afforded by classic".**
We have removed "the analyses afforded by classic".

**Lines 263-265: This is not clear, please, re-phrase it.**
We have rephrased this sentence to "Here we consider that $\Delta\alpha$, $\Delta f(\alpha)$ or by $\tau''(1)$ have equal weightings that reflect the overall multifractality of the data from the study area. As such, the ordering of these elements by $\Delta\alpha$, $\Delta f(\alpha)$ or by $\tau''(1)$ involved the summation of these values with the summed ordering then sorted again to compare the overall multifractality of these data".

**Line 267: I would change "parameters and coefficient of variation values" to "and basic statistic indices".**
We have changed "parameters and coefficient of variation values" to "and basic statistic indices" in the title of Table 3.

**Table 3: Using standard deviation is of not very much use here since its effect would depend on the magnitude of your data. For instance Zn has a very high standard deviation compared to Hg, so logically, this index would give Zn always the first order. I suggest only using coefficient of variation, since this index is normalized for all elements.**
We thank the reviewer for their suggestion and have revised Table 3, which is now only sorted by coefficient of variation values.

**Line 273: According to table 3, it should be "Hg>As>Zn>Cd>Pb>Cu".**
We have corrected this in the text to "Hg>As>Zn>Cd>Pb>Cu".

**Line 280: Why not performing this inverse distance weighting interpolation for the rest of the elements? Besides, be careful since here you are not using multifractals; however, your discussion is oriented as if this technique was multifractal.**
We wanted to use inverse distance weighting (IDW) interpolation to show that the multifractal spectrum technique and associated parameters can used to evaluate the degree of multifractality caused by human activities. The IDW interpolation is not the key focus of this paper and we have only used IDW to compare the spatial distribution of the elements with highest and lowest degrees of multifractality in the Daxing and Yicheng areas.

**Lines 284-285: Not exactly, only in the area where is a bunch of breeding facilities. In the case of Pb, this concentration would depend on the type of industry involved.**

We have calculated a correlation matrix for the density map of breeding factories in the Yicheng area (Fig. 7), and a filled contour map for Hg (Fig. 5). The result indicates a strong spatial correlation between the Hg contamination and the location of breeding facilities in the Yicheng area.

**Lines 287-290: What are they using as Hg source? It must appear from somewhere!!!**

We think fertilizer, fodder, pesticides and water could all be Hg sources in the Yicheng area, although further work is needed to identify the main source of contamination; this is beyond the scope of the current paper although we have mentioned this in the text.

**Lines 293-295: Please, re-phrase. This is not clear.**

We have rewritten this as follows "Here, we generated a correlation matrix that compares the relationship between the spatial density of breeding locations in the Yicheng area (Fig. 7) and filled contours maps showing the distribution of Hg (Fig. 5) and Cu (Fig. 6) in this region to identify whether there are any spatial correlations between the location of agricultural facilities and areas containing soils with elevated heavy metal concentrations (Table 4). The correlation matrix shows a significant correlation between agricultural facilities and high concentrations of Hg (correlation value = 0.434), whereas the location of these agricultural breeding facilities and areas of high Cu concentrations either have no relationship or are negatively correlated (correlation value = -0.064). This indicates that very little Cu has been anthropogenically added (or removed) from the soils in the Yicheng area, suggesting that these soils may contain only natural background concentrations of Cu and that the breeding activity in this area does not produce significant Cu contamination"

**Line 296: "a significant" instead of "an significant".**

We have used "a significant" instead of "an significant"

**Lines 297-299: I do not see this from your figure. In fact, the evolution of both elements is very similar at lower classes.**

We have accepted the reviewer's suggestion and have used a correlation matrix to show the correlation instead of a figure.

**Lines 301-302: I do not totally agree. You can see that the shape of the curve is similar for Cu and Hg, only different for the greater classes.**

Although the curves are similar, the new correlation matrix obtained during this study indicates a significant correlation between the location of agricultural facilities and high concentrations of Hg (correlation value = 0.434), whereas there is an independent or a negative correlation between agricultural breeding facilities and areas of high Cu concentrations (correlation value = -0.064).

**Line 304: "can efficiently reflect the multifractality", of course, they are designed to do this.**

We have removed this sentence to make this paragraph more brief.

**Line 305: Remove "by".**

We have removed "by".

**Lines 317-319: This is unclear. Please, re-phrase it.**

We have re-phrased this as follows: "Although somewhat beyond the scope of this study, the multi-element nature of the contamination in the Daxing area means that physical and chemical approaches to remediation (i.e., soil removal, soil vitrification, soil consolidation, electroremediation, or soil washing) are probably well suited for the remediation of heavy metal contaminated soil in this region (especially Pb).".

**Line 320: Remove "especially in areas with significant heavy metal pollution". From my viewpoint, it is not needed.**

We have removed "especially in areas with significant heavy metal pollution".

**Line 324: Remove "in this area".**

We have removed "in this area".

**Figure 4: In the upper left-hand side of the map there are some industries and the Pb concentration is rather low. Similar values are observed in the lower left-hand side of the map. This may indicate that Pb concentrations in soils depend more on the type of industry than on the fact that there is an industry, as you imply in your discussion.**

Figure 4 shows that only 4 factories are located in the area with low Pb concentrations in the upper and lower left hand sides of the maps, with all of the other factories located within areas with relatively high Pb concentrations (>34.61 mg/kg). As such, we agree with the reviewer although we still think that our data support our conclusion that areas with elevated Pb concentrations within the Daxing area are correlated to the location of industrial factories. However, we have also added a comment that reflects this variability as suggested by the reviewer.

**Figure 5: In contrast with the former figure, in the right-hand side of this map, we can observe high concentrations of Hg in the soil, but there are no breeding facilities on this part of the map… Then, why do these high Hg concentrations appear? According to your discussion, there is a direct relationship between the existence of a breeding facility and the high concentrations of Hg observed.**

**Figure 6: Coinciding with the former map, there are very high Cu concentrations in the right-hand side of this map, why? What is over there?**

Figure 5 indicates that some areas with the highest concentrations of Hg are spatially correlated with the breeding facilities, whereas other areas with slightly elevated concentrations of Hg are spatially correlated with the river in the right-hand side of the figure. In comparison, Figure 6 shows that a significant number of areas with elevated concentrations of Cu are located beside the river. We have discussed these variations within the text.

**Figure 7: To me, this graph is difficult to interprete. Both lines show correlations between number of facilities and Hg or Cu concentrations in soils. Besides, the caption is not clear. You talk about an anti-correlation and this is not observed in the graph.**

We have used a correlation matrix (Table 4) to show the relationship between the spatial density of breeding facilities in the Yicheng area and filled contours maps for the heavy metal concentrations in this area to show the correlation between these facilities and Hg concentrations and the lack of a correlation between these facilities and Cu concentrations in this region.

**Conclusions:**

**Line 355: According to table 3, the overal multifractality in Yicheng decreased as Hg>As>Zn>Cd>Pb>Cu and not as Hg>Zn>As>Cd>Pb>Cu as is reported here.**

We have revised this.

**Lines 356-364: What about other problems caused by the other heavy metals? People in Daxing are inmune to heavy metal pollution?**

The multi-element heavy metal pollution within the Daxing area is strongly influenced by human activities. However, the heavily industrial Daxing region contrasts with the highly agricultural nature of the Yicheng region, an area that supplies significant amounts of food to the city of Hefei. The contamination (especially Hg) in this area can more easily make its way into the human food chain (as well as being concentrated up the food chain), thereby having a more rapid direct impact on human health. As such, we have highlighted this in our paper to show that although both areas require remediation, the Hg contamination in the Yicheng area is especially worrisome.

**Line 365: "The initial results", I do not understand why you termed your results as "initial".**

We think that further research in this area is needed to identify the sources of pollution as mentioned by the reviewer earlier in their comments. As such, this paper is based in initial research that highlights areas for future research as well as demonstrating the validity of this multispectral approach. However, we acknowledge that the term "initial" may be confusing, and we have removed it.

**Lines 366-367: "multifractal parameters can efficiently reflect the multifractality caused by industrial and agricultural activities", well, of course, multifractal indices are designed to do this.**

We have revised this sentence to "The results presented here indicate that multifractal modeling and the associated three multifractal parameters ($\Delta\alpha$, $\Delta f(\alpha)$ and $\tau''(1)$) can efficiently reflect the multifractality caused by industrial and agricultural activities in the Daxing and Yicheng areas, respectively".

**Lines 369-370: "and the identification of major sources of heavy metal contamination". I do not agree, the identification of these sources was made through inverse distance weigthing interpolation, which is not a multifractal technique.**

We have revised this sentence to make it clearer as follows "This in turn indicates that multifractal modeling can be a useful approach in the evaluation of heavy metal pollution in soils and the identification of problematic heavy metals that need remediation in the research area.".

**RC2: Miranda, Jose  vivasm@gmail.com**

**Lines 234-240**

**The authors suggest a causal association between f(alfa) spectrum properties and contaminations. But the only real outcome of f(alfa) properties are related to heterogeneity and, as far as I know, there is no direct association between heterogeneity and contamination. The suggestion proposed by authors has no meaning. Unless the authors prove a correlation between heterogeneity and contamination.**

We thank the reviewer for their comment. The fact that the association between heterogeneity and contamination is hard to express means that we have removed the word "heterogeneity" from the paper.

**Lines 293-298**

**The authors answer:**

**Curves in figure 7 seems to be wrong, once in the map there are only 11 facilities and the graph shows 12. I can't understand how the graphs in Figure 7 can prove the existence of correlations between the location of industries and contamination. In my opinion, the fact that the number of facilities increase with rank is an obvious result, since a lower level contain all facilities of the highest levels. Thus, Figure 7 does not support the hypothesis of correlation neither causality between factors. I still think that statistical correlations could help to understand this relation.**

One of the breeding facilities is close to the margin of the figure and was clipped; however, we have revised this figure and more clearly demonstrated the correlation between the location of these facilities and heavy metal contamination using a statistical correlation (a correlation matrix) rather than using a figure.

**Technical Comments**

**Format problems in many equations (ex.: 2, 5, 7 and 8)**

We have revised equation 2.

We thank the anonymous referee and J. Miranda for their positive comments and have improved the written English and revised the confusing sentences within our paper. We hope that this manuscript is now acceptable for publication with the corrections and edits noted above. Please do not hesitate to contact me if you need any more information on or clarification of these revisions.

Yours faithfully,

Feng Yuan

[revised manuscript text omitted]

---

## Author Response (AR3)

Dear Prof. Schertzer and Dr. Miras-Avalos,

We thank these careful and constructive reviews of our manuscript. We have uploaded our response as a supplement to the comments and have incorporated these changes to our revised manuscript. For clarity, we have used a blue font for the reviewer's text, a black font for our text. We hope that after these revisions our manuscript will be considered suitable for publication in Nonlinear Processes in Geophysics.

**Abstract:**

**Line 22: "city. We used a multifractal" instead of "city and use a multifractal".**

We have used "city. We used a multifractal" instead of "city and use a multifractal".

**Line 25: Remove "for these soil geochemical data".**

We have removed "for these soil geochemical data".

**Lines 28-31: Please, consider re-phrasing this sentence. For instance, I propose to reduce it to "The degree of multifractality suggests that the differing economic activities in Daxing and Yicheng generate very different heavy metal pollution loads".**

We agree to this suggestion and re-phrasing this sentence.

**Introduction and overview of the study area:**

**Line 47: "may allow for assessing" instead of "can investigate".**

We have used "may allow for assessing" instead of "can investigate".

**Line 48: "with pollutants, as well as" instead of "with pollutants and as well as".**

We have used "with pollutants, as well as" instead of "with pollutants and as well a".

**Line 61: Remove "in this case".**

We have removed "in this case".

**Lines 72-73: Please, check the order of citations here. Besides, please, consider reducing the number of citations seems I think there are far too many than necessary.**

We have reduced some citations here.

**Study area and geochemical data:**

**Line 109: Remove "was" from "was monitored".**

We have removed "was" from "was monitored".

**Multifractal spectrum analysis:**

**Check the readability of equation 1. Parenthesis and symbols are overlapping.**

It is a format conversion error. We have used Adobe acrobat to convert the ms-doc file to pdf file again. Now it is correct.

**Lines 141-142: You should indicate what are $\alpha$ and $f(\alpha)$.**

We have updated this sentence to "The values of $\tau(q)$ derived using this equation can be then used to determine singularity $\alpha$ and fractal spectra $f(\alpha)$ values using a Legendre transformation".

**Line 146: "a given dataset" instead of "the dataset in question". Indicate the symbols that you used for "left and right branches".**

We have used "a given dataset" instead of "the dataset in question", and indicated $\Delta\alpha_L$ and $\Delta\alpha_R$ in this sentence.

**Lines 148-150: Check the readability of equations 4 to 6. The "min" is not visible.**

It is a format conversion error. We used Adobe acrobat to convert the ms-doc file to pdf file again. Now it is correct.

**Lines 156-160: Re-phrase, not clear.**

We have rephrased this sentence to "In addition, local multifractality $\tau''(1)$, which may determined by ordinary spatial analysis functions (autocorrelations and semivariograms), can also be used as a measure to quantitatively characterize the multifractality of a dataset using equation 8".

**Line 164: "monofractal" instead of "single fractal".**

We have used "monofractal" instead of "single fractal".

**Lines 165-168: Consider removing, already stated in lines 77-80.**

We have removed the sentence between 77-80.

**Geochemical analysis results**

**Lines 173-174: Ok, but the minimum and the mean were higher in the Daxing area.**

The minimum of Cd in Yicheng area is a little bit higher than Daxing area, so in this sentence we didn't include the minimum in this sentence.

**Line 185: I do not think that you need multifractals to discriminate this.**

We have updated this sentence to "This means that multifractal techniques are highly suited for the characterization of the geochemistry of the soils".

**Calculation processes of multifractal spectrum and discussion**

**Lines 196-197: "using a range of q values from -10 to 10 with and interval of 1" should be stated in the "multifractal spectrum analysis" section.**

We have moved these words to the "multifractal spectrum analysis" section.

**Line 212: "have a high multifractality" instead of "have highly multifractality".**

We have used "have a high multifractality" instead of "have highly multifractality"

**Line 215: It should be $\varDelta\alpha$ instead of $\alpha$.**

We have revised this, used $\Delta\alpha$ instead of $\alpha$.

**Line 222: "but not maximum", if you look at your histogram (figure 2) you will see that most of the samples from Yicheng were below 0.31 mg/kg of Hg, whereas in Daxing there are more samples above 0.5 mg/kg of Hg.**

Thanks for this comment, we have removed the "but not maximum".

**Line 229: What do you mean by "deleterious effects such as the heavy metal pollution of people, crops and animals"?**

In order to make this point more clear, we have rephrase this sentence to "This means that although contamination in both areas needs to be evaluated further and should be remediated to avoid any deleterious effects, the fact that the Hg contamination in the Yicheng area may be more bioavailable and may have a larger effect on the population of this region (as a result of the agricultural activity in this area) means it should be considered a priority".

**Line 236: "the elements within the samples from" instead of "different elements within".**

We have used "the elements within the samples from" instead of "different elements within".

**Lines 238-245: Please, remove all references to standard deviation because it is not used for sorting the elements according to table 3.**

We have revised these lines as follows: "The data shown in Table 3 indicates that the Pb data within the Daxing area has close to the lowest coefficient of variation, but largest the $\Delta f(\alpha)$ and $\tau''(1)$ values for these Pb data are indicative of strongest multifractality compared to the other heavy metals in the soils within the Daxing area. In comparison, the As data for soils in the Daxing area yielded the largest coefficient of variation but the moderate $\Delta f(\alpha)$ and $\tau''(1)$ values, indicating these As data only have moderate multifractality. These differences indicate that the multifractal parameters reveal new information about the nonlinear variability and the characteristics of these geochemical data compared to the basic statistics for these samples. ".

**Line 262: "indicate" instead of "indicates".**

We have used "indicate" instead of "indicates".

**Line 276: "polluting more than others" instead of "more polluting than others".**

We have used "polluting more than others" instead of "more polluting than others".

**Lines 296-297: How was spatial density assessed? It is not explained anywhere.**

We have explained it in the title of Fig. 7 as follows: "generated using the Kernel Density method within spatial analyst tools of the ArcGIS software package)".

**Lines 302-305: When you are talking about correlation value, do you refer to coefficient of correlation? Could you indicate the p-values for these correlations, please?**

Yes, we have used "coefficient of correlation" instead of "correlation value" in these lines and the title of Table. 4. We are sorry that ArcGIS software can not provide p-values for the correlation matrix, but we think the coefficient of correlation can be used to compare the correlation between two layers very well.

**Line 308: "do" instead of "does".**

We have used "do" instead of "does".

**Lines 308-311: Please, re-phrase, not clear.**

We have rephrased this sentence as follows: "The negative correlation coefficient, symmetrical distribution and weakest multifractality of Cu give one clue to the spatial relationship between Cu contamination and the river in the right hand side of Fig. 6."

**Line 339: Include a parenthesis after "water".**

We have revised this sentence to " In addition, the source of the Hg contamination (e.g. fertilizer, fodder, pesticides, water, or some other source) remains unclear."

**Line 340: Remove "or other source".**

We have removed "or other source".

**Line 346: Inverse distance weighted interpolation is not mentioned or explained in the methodology.**

The Inverse distance weighted interpolation is an well known interpolation method. In this paper, we used this method provided by ArcGIS software to interpolate the data. Because this method is not the key point of this paper, we revised the title of Fig. 4~ Fig. 6 to show the method and software we have used. If the reader want to know the detail about this method, they can find it in the manual of the ArcGIS software package.

**Line 360: How was this density map created? This should be explained in the methods.**

We have revised the title of Fig. 7 to show the method and software we have used.

**Table 4: How was this correlation matrix made? It is not explained in the methodology.**

We have revised the title of Table 4 to show the method and software we have used.

**Lines 365-367: There is no figure 8.**

We have revised it.

**Lines 368-371: Is this needed?**

We have removed these sentences.

**Conclusions:**

**Lines 374-376: Consider removing the first sentence.**

We have removed the first sentence.

**Line 382: Remove "the overall order in soils".**

We have removed "the overall order in soils".

**Line 390: Remove the word "well".**

We have removed "well" from this sentence.

**Lines 393-396: I think that this sentence can be removed.**

We have removed this sentence.

We thank again for these positive comments and helping us revising the confusing sentences within our paper. We hope that this manuscript is now acceptable for publication with the corrections and edits noted above. Please do not hesitate to contact me if you need any more information on or clarification of these revisions.

Yours faithfully,
Feng Yuan

[revised manuscript text omitted]